# Equivariant Covariance Tensors: Guaranteed SPD Uncertainty for Tensor-Valued Geometric Learning

**Ruihan Liu** [1]  **Yu Ji** [1]  **Jianbo Yu** [2]  **Shifu Yan** [3]  **Qingchao Jiang** [4]

## Abstract

Tensor-valued prediction is fundamental to geometric deep learning, yet uncertainty quantification (UQ) for such outputs remains an open challenge. While E(3)-equivariant neural networks excel at point estimates, they lack rigorous confidence measures. We focus on symmetric rank-2 tensor prediction, where the target has six Kelvin–Mandel coordinates and full uncertainty is represented by a $6 \times 6$ covariance matrix. We introduce a framework for E(3)-equivariant UQ, modeling the full predictive distribution where both mean and covariance preserve rotational symmetry. Our approach decomposes the covariance into irreducible representations $\mathrm{Sym}^2(\rho_c) \cong 2 \times (l = 0) \oplus 2 \times (l = 2) \oplus 1 \times (l = 4)$. By mapping from the flat Lie algebra $\mathfrak{sym}(6)$ to the curved SPD manifold via matrix exponentiation, we strictly ensure positive-definite covariances while maintaining exact equivariance. Furthermore, we formulate a Log-Euclidean Equivariant Scoring Objective (LE-ESO)—a robust surrogate loss based on the Multivariate Laplace distribution—providing robustness to heavy-tailed errors and stable optimization. Validation on ModelNet40 inertia tensors and Materials Project dielectric tensors demonstrates that our method achieves competitive performance and provides physically consistent, symmetry-preserving uncertainty estimates with useful risk and OOD sensitivity.

[1]College of Intelligent Robotics and Advanced Manufacturing, Fudan University, Shanghai 200433, China [2]School of Microelectronics, Fudan University, Shanghai 200433, China [3]ByteDance, Beijing, China [4]School of Information Science and Engineering, East China University of Science and Technology, Shanghai 200237, China. Correspondence to: Jianbo Yu <jb_yu@fudan.edu.cn>.

*Proceedings of the 43rd International Conference on Machine Learning*, Seoul, South Korea. PMLR 306, 2026. Copyright 2026 by the author(s).

## 1. Introduction

Tensor-valued predictions are fundamental to scientific computing and geometric deep learning, with applications spanning material properties (elasticity tensors, dielectric response), biomedical imaging (diffusion MRI), and computational fluid dynamics. While E(3)-equivariant neural networks (ENNs) have achieved remarkable success in predicting tensorial properties like dielectric constants (Batatia et al., 2022; Heilman et al., 2024), they are inherently deterministic—outputting single point estimates without any confidence measure. This is a critical limitation: overconfident tensor predictions in scientific decisions can lead to costly experimental failures.

In this paper, the primary output is a symmetric rank-2 tensor $C \in \mathbb{R}^{3 \times 3}_{\mathrm{sym}}$, such as a dielectric tensor. Although $C$ is a $3 \times 3$ matrix, symmetry leaves six independent degrees of freedom. We represent it by the Kelvin-Mandel vector $\mathbf{c} \in \mathbb{R}^6$, for which rotations act by an orthogonal six-dimensional representation $\rho_c(R)$. Therefore, uncertainty over $C$ is not a $3 \times 3$ covariance, but a $6 \times 6$ covariance over the Kelvin-Mandel coordinates.

Extending ENNs with uncertainty quantification poses a fundamental challenge: uncertainty estimates must themselves transform equivariantly. Mathematically, the covariance $\Sigma \in \mathbb{R}^{6 \times 6}$ must satisfy:

$$\Sigma(R \cdot X) = \rho_c(R)\Sigma(X)\rho_c(R)^\top, \quad \forall R \in O(3),$$

where $\rho_c(R)$ is the rotation matrix in Kelvin-Mandel notation. This creates a parameterization challenge: Cholesky-type covariance heads guarantee SPD but are not equivariant in Kelvin–Mandel covariance coordinates, while direct equivariant regression preserves the transformation law but does not guarantee SPD.

In this work, we introduce a principled framework for E(3)-equivariant full-covariance uncertainty quantification in symmetric tensor-valued prediction. Our contributions address this equivariant SPD parameterization challenge through: (1) an equivariant matrix-exponential head parameterizing the SPD manifold via irreducible decomposition $\mathrm{Sym}^2(\rho_c) \cong 2 \times (\ell = 0) \oplus 2 \times (\ell = 2) \oplus 1 \times (\ell = 4)$; (2) a stable joint optimization strategy via a Log-Euclidean

scoring objective that integrates uncertainty calibration with geometric feature extraction; and (3) rigorous validation on ModelNet40 inertia tensors with competitive performance on Materials Project dielectric prediction (MAE 1.55).

## 2. Related Work

**Equivariant Tensor Prediction.** E(3)-equivariant neural networks (ENNs) achieve state-of-the-art tensor prediction across materials science and 3D geometry (Batatia et al., 2022; Heilman et al., 2024; Hua et al., 2026; Pakornchote et al., 2023; Fung et al., 2021; Reiser et al., 2022; Du et al., 2024; Equer et al., 2023). However, all existing ENNs are inherently deterministic—they cannot quantify confidence in their predictions. Our framework preserves equivariance guarantees while adding rigorous uncertainty quantification, addressing this critical gap. While our approach utilizes the spherical harmonic basis provided by e3nn (Geiger & Smidt, 2022), recent advances in representation theory, such as the High-Rank Irreducible Cartesian Tensor (ICT) decomposition framework (Shao et al., 2025), offer alternative analytical paths for constructing equivariant bases directly in Cartesian space. Such methods could potentially simplify the implementation for higher-rank tensor properties.

**SPD Constraints in Neural Networks.** Ensuring symmetric positive-definite (SPD) covariances is essential for valid uncertainty. Prior work uses Cholesky decomposition, eigendecomposition, or Riemannian optimization (Jekel et al., 2022; Pouliquen et al., 2025; Zhao et al., 2023). These methods guarantee SPD in a fixed coordinate parameterization, but they do not by themselves provide an equivariant map $X \mapsto \Sigma(X)$ under the covariance representation $\rho_c$. Orthogonal conjugation preserves SPD; thus the difficulty is not a conflict between the SPD property and rotation itself. Rather, the challenge is to design a neural parameterization that is simultaneously equivariant and constrained to the SPD cone. Cholesky-type heads enforce SPD but are not equivariant in Kelvin-Mandel covariance coordinates, whereas direct equivariant regression can preserve the transformation law but does not guarantee SPD. Our matrix-exponential head resolves this parameterization problem by predicting an equivariant symmetric operator $A(X)$ and setting $\Sigma(X) = \exp(A(X))$, which satisfies $\exp(\rho_c(R)A\rho_c(R)^\top) = \rho_c(R)\exp(A)\rho_c(R)^\top$ and therefore is jointly SPD and equivariant in all rotated frames.

**Probabilistic Methods and Equivariant GPs.** Probabilistic extensions for ENNs are severely underdeveloped. While ensemble methods can provide heuristic uncertainty estimates (Rudner et al., 2022) and even exhibit emergent equivariance (Gerken & Kessel, 2024), our framework explicitly learns the full 21-parameter aleatoric covariance tensor, which is essential for modeling the inherent anisotropic

noise in physical properties. Bayesian neural networks face computational challenges at scale (Rensmeyer et al., 2024; Olivier et al., 2021; Doan et al., 2025; Sheinkman & Wade, 2025). Existing equivariant Bayesian approaches focus on scalar or vector quantities (Zhou et al., 2024), missing tensor-valued uncertainty. E(3)-equivariant Gaussian processes provide theoretically sound uncertainties but are typically restricted to isotropic or diagonal covariance approximations (Steinert et al., 2025; Bevanda et al., 2025)—scaling them to the full 21-parameter covariance structure of rank-2 tensors remains computationally prohibitive. Our neural network approach offers computational scalability while learning complete equivariant tensor correlations. Recent work on uncertainty calibration and stable tensor operations provides theoretical grounding for our design (Berman et al., 2026; Gruber & Buettner, 2022; Fakour et al., 2024; Newman et al., 2024).

## 3. Methods

### 3.1. Problem Formulation

We address the fundamental challenge of predicting tensor-valued quantities while quantifying predictive uncertainty. We present the construction for symmetric rank-2 tensors $C \in \mathbb{R}^{3\times3}_{\text{sym}}$, which already require a full $6 \times 6$ covariance representation. The SPD construction and scoring objective are representation-agnostic once an equivariant symmetric operator $A(X)$ is available. However, the parameterization of $A(X)$ is representation-specific and must be constructed separately for each tensor order and symmetry group. We predict $C$ from a 3D structure $X = \{(\mathbf{x}_i, f_i)\}_{i=1}^N$, where $\mathbf{x}_i \in \mathbb{R}^3$ denotes spatial coordinates and $f_i$ represents associated features (such as atomic species in materials). This formulation encompasses important applications such as predicting dielectric tensors in crystal structures. Although we demonstrate this framework on materials science, the approach is applicable to any 3D point cloud with rank-2 tensorial attributes, ranging from biological molecules to geometric shapes.

Rather than producing deterministic point estimates, we model the full predictive distribution

$$p(C \mid X) = \text{Laplace}(C \mid \mu(X), \Sigma(X)), \qquad (1)$$

where $\mu(X) \in \mathbb{R}^{3\times3}_{\text{sym}}$ is the predicted mean tensor and $\Sigma(X)$ captures the predictive uncertainty through a covariance structure.

To maintain geometric consistency across all domains, both the mean prediction $\mu$ and covariance $\Sigma$ must satisfy equivariance constraints with respect to rotations and reflections. Since we predict global tensor properties (e.g., dielectric tensor of a unit cell), the predictions are invariant to translations

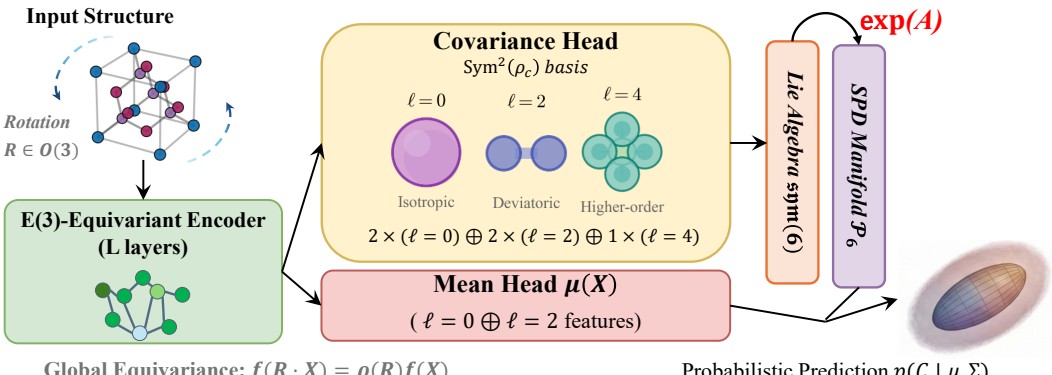

*Figure 1.* **E(3)-equivariant tensor uncertainty framework.** (1) Feature Decomposition: covariance head models symmetry via irreducible representations ($\ell = 0, 2, 4$). (2) Manifold Mapping: network predicts unconstrained $A \in \mathfrak{sym}(6)$. (3) Validity Constraint: $\Sigma = \exp(A)$ projects to SPD manifold $\mathcal{P}_6$.

but equivariant to orthogonal transformations:

$$\mu(R \cdot X) = R\mu(X)R^\top,$$
$$\Sigma(R \cdot X) = \rho_c(R)\,\Sigma(X)\,\rho_c(R)^\top. \tag{2}$$

Our construction ensures full $O(3)$ equivariance. For symmetric rank-2 tensors, the transformation under reflection ($\det R = -1$) is handled by the even-parity representation of the Kelvin-Mandel basis $\rho_c(R) = R \otimes_s R$, where $(\det R)^2 = 1$ guarantees consistent behavior for both chiral and achiral structures. We numerically verify $O(3)$ equivariance in Section 4.5 (Table 4), with detailed analysis in Appendix E.3. Predictions are translation-invariant since global tensor properties depend only on relative atomic positions. This universal equivariance constraint forms the foundation of our domain-agnostic framework. We emphasize that while Eq. 1 presents the distribution in standard predictive form, the training optimizes the robustified LE-ESO objective (Section 3.5) which generalizes the Multivariate Laplace negative log-likelihood with enhanced stability against outliers.

### 3.2. Voigt Representation and Covariance Structure

To facilitate neural network implementation while preserving the tensor's geometric structure, we employ the Kelvin-Mandel notation to flatten the symmetric tensor $C$ into a 6-dimensional vector. Unlike standard Voigt representation, Kelvin-Mandel notation maintains the isometry property between tensor and vector spaces:

$$\mathbf{c}_{\text{KM}} = [C_{11}, C_{22}, C_{33}, \sqrt{2}C_{23}, \sqrt{2}C_{13}, \sqrt{2}C_{12}]^\top, \tag{3}$$

which preserves the Frobenius norm: $\|\mathbf{c}_{\text{KM}}\|_2 = \|C\|_F$. Under rotation $R$, $\mathbf{c}'_{\text{KM}} = \rho_c(R)\mathbf{c}_{\text{KM}}$ where $\rho_c(R)$ is an orthogonal $6 \times 6$ matrix. The covariance transforms as $\Sigma' = \rho_c(R)\Sigma\rho_c(R)^\top$, maintaining coordinate invariance of the physical uncertainty.

### 3.3. Irreducible Representation Decomposition

The mathematical structure of equivariant uncertainty quantification becomes clear through representation theory. In group theory, irreducible representations are fundamental building blocks that cannot be further decomposed into smaller invariant subspaces. The 6D representation $\rho_c$ for symmetric $3 \times 3$ tensors decomposes into irreducible representations of SO(3) as

$$\rho_c \cong l = 0 \oplus l = 2, \tag{4}$$

corresponding respectively to the isotropic (trace) and deviatoric (traceless) components of the tensor. Here $\cong$ denotes an isomorphism of representations, not equality of matrices: after a fixed change of basis, the six Kelvin-Mandel coordinates split into a one-dimensional isotropic trace component and a five-dimensional traceless deviatoric component.

Since $\Sigma$ transforms as $\rho_c \otimes \rho_c$, its representation decomposes as:

$$\rho_c \otimes \rho_c = (l = 0 \oplus l = 2) \otimes (l = 0 \oplus l = 2)$$
$$= (l = 0) \oplus 2(l = 2) \oplus (l = 4) \oplus (l = 1, 3)_{\text{antisym}}. \tag{5}$$

The $\ell = 1$ and $\ell = 3$ components lie in the antisymmetric part of $\rho_c \otimes \rho_c$, corresponding to operators that change sign under exchange of the two covariance indices. Since covariance matrices satisfy $\Sigma = \Sigma^\top$, only the symmetric square $\text{Sym}^2(\rho_c)$ remains, yielding:

$$\text{Sym}^2(\rho_c) \cong 2 \times (l = 0) \oplus 2 \times (l = 2) \oplus 1 \times (l = 4), \tag{6}$$

This provides 21 independent degrees of freedom for symmetric $6 \times 6$ SPD covariance matrices.

### 3.4. Equivariant Neural Architecture

Figure 1 summarizes the data flow. A shared E(3)-equivariant encoder first maps the input structure $X$ to latent irreducible features. The mean head selects the $\ell = 0 \oplus \ell = 2$ components and outputs the Kelvin-Mandel mean vector $\boldsymbol{\mu}_{\mathrm{KM}}(X) \in \mathbb{R}^6$, which is mapped back to a symmetric $3 \times 3$ tensor. The covariance head outputs coefficients in $2 \times (\ell = 0) \oplus 2 \times (\ell = 2) \oplus 1 \times (\ell = 4)$, which are assembled into an equivariant symmetric operator $A(X) \in \mathfrak{sym}(6)$. Finally, the predictive covariance is $\Sigma(X) = \exp(A(X))$, and the loss is computed from the Kelvin-Mandel residual $\Delta \mathbf{c} = \mathbf{c}_{\mathrm{KM}} - \boldsymbol{\mu}_{\mathrm{KM}}(X)$.

The covariance head $f_\Sigma : X \mapsto A(X)$ must satisfy the transformation property

$$A(R \cdot X) = \rho_c(R) A(X) \rho_c(R)^\top. \tag{7}$$

Following the irreducible representation decomposition in Eq. 6, we construct $A(X)$ through structured Clebsch-Gordan combinations. Let $\phi^{(L)}(X) \in \mathbb{R}^{F_L \times (2L+1)}$ denote spherical tensor features of order $L$ produced by the equivariant backbone. We index the irreps appearing in $\mathrm{Sym}^2(\rho_c)$ by

$$\mathcal{I} = \{(0,1), (0,2), (2,1), (2,2), (4,1)\},$$

where $(L, r)$ refers to the $r$-th copy of the $L$-irrep, and write

$$A(X) = \sum_{(L,r) \in \mathcal{I}} \sum_{m=-L}^{L} a_{L,m}^{(r)}(X) B_{L,m}^{(r)}, \tag{8}$$

with the equivariant coefficients

$$a_{L,m}^{(r)}(X) = \sum_{f=1}^{F_L} w_{L,r,f} \, \phi_{f,m}^{(L)}(X).$$

Here $B_{L,m}^{(r)} \in \mathbb{R}_{\mathrm{sym}}^{6 \times 6}$ are *fixed, input-independent* basis matrices for the $r$-th copy of the $L$-irrep in $\mathrm{Sym}^2(\rho_c)$, computed once from the Clebsch-Gordan decomposition. All input dependence resides in the equivariant coefficients $a_{L,m}^{(r)}(X)$. Under rotation, the coefficient vector $(a_{L,m}^{(r)})_{m=-L}^{L}$ and the basis $(B_{L,m}^{(r)})_{m=-L}^{L}$ transform by the same Wigner-$D^{(L)}$ representation, so their contraction yields $A(R \cdot X) = \rho_c(R) A(X) \rho_c(R)^\top$. This explicit tensor basis construction guarantees that any choice of weights $w_{L,r,f}$ preserves the equivariance property, providing hard-constrained geometric consistency rather than soft-regularized approximation. We note that while we rely on spherical tensor products, the orthogonal ICT decomposition matrices (Shao et al., 2025) provide an equivalent and highly efficient basis

for higher-order Cartesian tensors, which may offer computational advantages for future extensions to rank-4 tensors like elasticity.

Our architecture implements an E(3)-equivariant neural network using the `e3nn` library, following the established paradigm of equivariant message passing. The implementation proceeds through two key stages. First, an equivariant message passing backbone processes the atomic structure to produce latent features transforming under mixed irreps up to $\ell_{max} = 4$. Second, an equivariant linear layer—implemented via a fourth-order Cartesian tensor with symmetry $ijkl = jikl = ijlk = klij$—assembles these latent features into the symmetric block structure of $A$, ensuring consistency with the $\mathrm{Sym}^2(\rho_c)$ representation while automatically filtering to the $l = 0, 2, 4$ components required for rank-4 covariance output. The covariance head uses a residual connection $A = A_{\mathrm{base}} \cdot I + \Delta A$, where $A_{\mathrm{base}}$ provides an isotropic baseline and $\Delta A$ captures the anisotropic uncertainty structure learned from the data.

We employ an end-to-end joint optimization strategy, where both the mean and covariance heads are trained simultaneously. The inherent stability of our matrix-exponential mapping and the LE-ESO loss eliminates the need for gradient detachment, allowing the backbone to learn geometric features that are mutually informative for both point prediction and uncertainty quantification. This design guarantees that the raw network output $A$ naturally possesses the correct equivariance properties, setting the stage for positive-definite covariance construction.

### 3.5. Positive-Definite Covariance Construction and Training Objective

**From Curved Manifold to Flat Tangent Space.** To strictly enforce the SPD constraint while maintaining equivariance, we leverage the Log-Euclidean framework (Arsigny et al., 2006) and parameterize $\Sigma(X)$ via the matrix exponential mapping:

$$\Sigma(X) = \exp(A(X)), \tag{9}$$

We thus optimize within the tangent space $\mathfrak{sym}(6)$—a flat Euclidean vector space at the identity—and the matrix exponential lifts $A(X)$ to the SPD manifold $\mathcal{P}_6$ while preserving E(3)-equivariance, since $\rho_c(R)$ acts by orthogonal conjugation (Figure 2).

**Log-Euclidean Equivariant Scoring Objective (LE-ESO).** Given the Kelvin-Mandel residual $\Delta \mathbf{c} = \mathbf{c}_{\mathrm{KM}} - \boldsymbol{\mu}_{\mathrm{KM}}(X)$, we define the Log-Euclidean Mahalanobis distance

$$D_M(A, \Delta \mathbf{c}) = \sqrt{\Delta \mathbf{c}^\top \exp(-A) \Delta \mathbf{c}}. \tag{10}$$

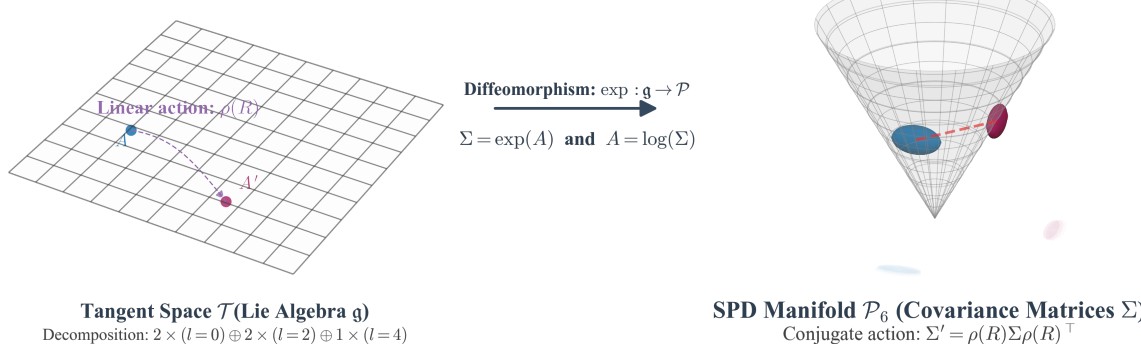

**Tangent Space $\mathcal{T}$ (Lie Algebra $\mathfrak{g}$)**
Decomposition: $2 \times (l=0) \oplus 2 \times (l=2) \oplus 1 \times (l=4)$

**SPD Manifold $\mathcal{P}_6$ (Covariance Matrices $\Sigma$)**
Conjugate action: $\Sigma' = \rho(R) \Sigma \rho(R)^\top$

*Figure 2.* **Geometric interpretation of equivariant covariance construction.** The network operates in the Lie algebra $\mathfrak{g} \cong \mathbb{R}_{\text{sym}}^{6 \times 6}$ (left), producing $A(X)$. The exponential map $\Sigma = \exp(A)$ (center) projects onto the SPD manifold $\mathcal{P}_6$ (right), guaranteeing valid uncertainties while preserving $E(3)$-equivariance.

To control the influence of extreme outliers, we use a robustified distance $\tilde{D}_M$ with transition threshold $\tau$:

$$\tilde{D}_M = \begin{cases} D_M, & D_M < \tau, \\ \tau + \log(1 + D_M - \tau), & D_M \geq \tau. \end{cases} \quad (11)$$

The final LE-ESO objective combines uncertainty volume regularization with the (robustified) data-fit term:

$$\mathcal{L}_{\text{LE-ESO}} = \alpha \, \text{Tr}(A) + \tilde{D}_M, \quad (12)$$

where $\alpha > 0$ controls the trade-off between uncertainty volume and data fit.

When $\alpha = 1$ and $\tilde{D}_M = D_M$, this is the multivariate Laplace negative log-likelihood in Log-Euclidean form, a strictly proper scoring rule on $\mathfrak{sym}(6)$ (Gneiting & Raftery, 2007). With log-tail robustification or $\alpha \neq 1$, the objective is a *robust surrogate scoring objective*, trading strict propriety for outlier stability. We set $\alpha = 1$ in our primary experiments; Appendix D.4 reports a validation sweep over $\alpha \in \{0.03, 0.1, 0.3, 1.0\}$ showing stable training, with $\alpha = 1$ also giving the lowest validation MAE. Detailed derivation and gradient analysis are in Appendix A.4.

**Training Stability via Joint Optimization.** In our experiments, the Lie algebra parameterization combined with eigenvalue clamping and the log-tail robustification in Eq. 11 keeps gradients and the matrix exponential numerically well-behaved during training, enabling stable end-to-end joint optimization of both the mean and covariance heads without gradient detachment, so the backbone can receive informative gradients from the uncertainty quantification objective. This joint training paradigm allows the learned geometric features to be optimized for both prediction accuracy and uncertainty calibration, without sacrificing numerical stability or equivariance guarantees.

## 4. Experiments

We evaluate the proposed framework in two main settings: (i) controlled geometric validation on ModelNet40 inertia tensors (Wu et al., 2015), and (ii) real-data dielectric tensor prediction on the Materials Project (Barroso-Luque et al., 2024; Jain et al., 2013). These main experiments are complemented by additional studies in Appendix D, including ModelNet40 shape-covariance prediction (Appendix D.1), rank-4 elasticity tensor prediction (Appendix D.2), runtime profiling (Appendix D.3), and sensitivity to the LE-ESO weight $\alpha$ (Appendix D.4). Dataset statistics are detailed in Appendix C.1.

We compare our equivariant full-covariance model against two primary baselines: (i) a deterministic model trained with MSE, and (ii) a diagonal UQ model assuming independent components. For ablation analysis, we evaluate non-equivariant and non-SPD variants (Section 4.5). All estimators utilize an E(3)-equivariant backbone with $\ell_{max} = 4$ to support rank-4 covariance output; detailed hyperparameters and training protocols are provided in Appendix B.

Our experiments verify three central hypotheses: (1) geometric validation—exact equivariance preservation on complex 3D shapes; (2) accuracy—maintained point-prediction performance while modeling full covariance structure; and (3) calibration—covariance matrices that reflect true error distributions while respecting underlying geometry.

### 4.1. Controlled Geometric Validation: Inertia Tensor Prediction

The ModelNet40 experiments are intended as controlled geometric validation rather than as replacements for closed-form tensor estimators. For inertia tensors, analytic formulas exist; our goal is to isolate whether the learned covariance remains equivariant, SPD, and geometrically meaningful under controlled point-cloud perturbations. The inertia tensor

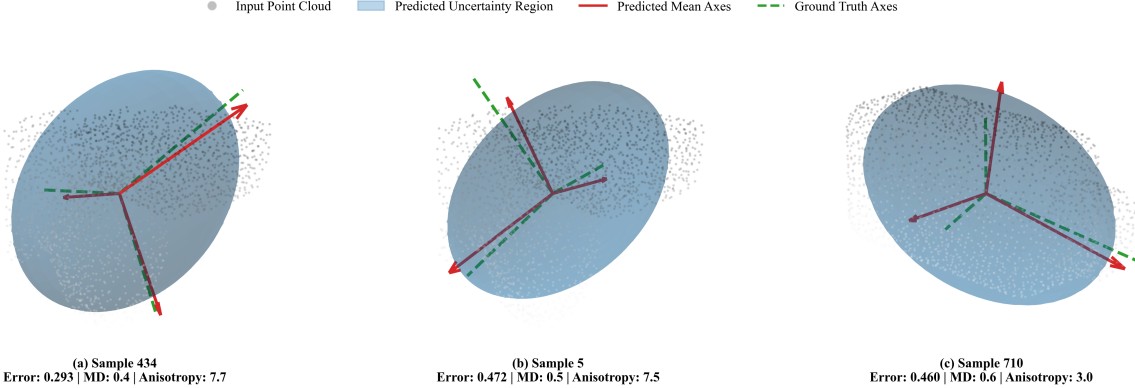

*Figure 3.* **3D uncertainty visualization on ModelNet40.** (Top) Point clouds with uncertainty ellipsoids (blue) and principal axes: predicted (red) vs ground truth (green). (Bottom) Anisotropic covariance matrices. Samples show varying errors (0.293–0.472) and Mahalanobis distances (0.37–0.50), demonstrating geometric-adaptive uncertainty that rotates with the object shape.

$\mathcal{I}$ transforms equivariantly under rotation ($\mathcal{I}' = R\mathcal{I}R^\top$), making it a clean target for verifying geometric consistency. We use the official split of 12,311 CAD models and introduce aleatoric uncertainty via Gaussian jitter applied to point positions (detailed preprocessing in Appendix C.1). For this validation task, the mean and covariance heads are trained jointly without gradient detachment, as the synthetic noise is well-behaved. We provide a second controlled rank-2 tensor validation on ModelNet40 shape covariance in Appendix D.1, showing that the construction is not tied to the inertia target.

We quantify equivariance error using the relative Frobenius norm $E_{\text{equiv}} = \|\Sigma(R \cdot X) - \rho_c(R)\Sigma(X)\rho_c(R)^\top\|_F / \|\Sigma(X)\|_F$. Prediction accuracy and uncertainty scores are reported in Table 1; equivariance and SPD-validity results are analyzed separately in Table 4, where the equivariance errors remain at the level of $10^{-7}$, confirming near-machine-precision symmetry preservation. Our full-covariance model reduces MAE by 15% relative to the diagonal baseline while maintaining perfect SPD properties ($>99.9\%$ validity, median condition number 6.8). Detailed SPD analysis is provided in Appendix E.5.

*Table 1.* ModelNet40 inertia tensor prediction. Our equivariant full-covariance framework achieves strong performance with exact geometric consistency and physical constraints.

| METHOD | MAE | RMSE | LE-ESO | SPD |
|---|---|---|---|---|
| OURS (FULL COV) | **0.078** | **0.128** | **10.66** | ✓ |
| DIAGONAL COV | 0.092 | 0.156 | 12.84 | ✓ |
| DETERMINISTIC (MSE) | 0.083 | 0.135 | N/A | ✗ |

Figure 3 provides visual validation that our uncertainty estimates are physically meaningful: uncertainty ellipsoids align with principal shape axes (demonstrating E(3)-equivariance), expand in regions with sparse point density

(capturing sampling ambiguity), and preserve tensorial correlations across components.

### 4.2. Application: Dielectric Tensor Prediction

**Dataset and Preprocessing.** We utilize the Materials Project dielectric tensor dataset (Barroso-Luque et al., 2024; Jain et al., 2013) with static dielectric tensors computed via DFPT. To ensure data quality and consistency, we apply systematic filtering criteria (detailed in Appendix C.1), including structure size constraints ($3 \leq$ atoms $\leq 30$), SPD positive-definiteness verification, and value range constraints. We apply Matrix Log-Normalization to reduce the dynamic range of dielectric tensors before converting them to Kelvin–Mandel coordinates, ensuring that the six independent components are scaled consistently across diverse crystal structures. We process atomic structures into graphs with $5.0$Å cutoff. Architecture and training details are provided in Appendix B.

### 4.3. Prediction Accuracy

Our goal is not state-of-the-art point prediction alone, but competitive accuracy with full-covariance, symmetry-preserving uncertainty. Table 2 shows our method achieves competitive MAE among UQ models (1.55, vs. 1.96 for the MACE deep ensemble and 2.25 for diagonal UQ), and remains close to deterministic point predictors such as DT-Net (Mao et al., 2024) (1.91) and GoeCTP (Hua et al., 2026) (1.41), which do not provide calibrated equivariant covariance estimates. The primary contribution is the principled, backbone-agnostic UQ mechanism, not a new point estimator. Modeling the full covariance manifold does not hinder mean estimation; the gains are most visible on off-diagonal components, which capture anisotropic directional dependencies.

Crucially, the parity plot in Figure 4a demonstrates that the

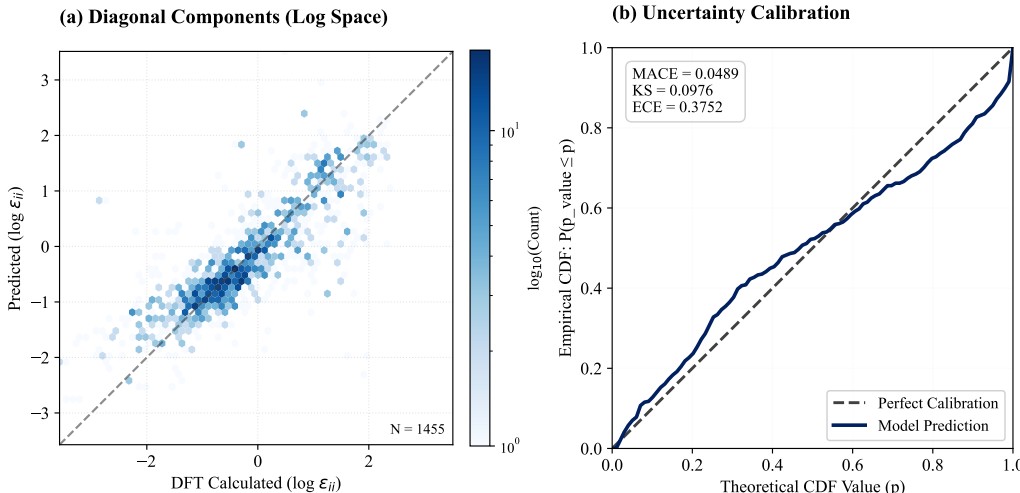

*Figure 4.* **Performance on Materials Project.** (a) Parity plot of diagonal components ($R^2 = 0.659$). (b) Multivariate Laplace reliability diagram (MACE= 0.0489).

*Table 2.* Performance on Materials Project dielectric dataset. GoeCTP (Hua et al., 2026) employs a scalable equivariant architecture. DTNet (Mao et al., 2024) uses universal potential embeddings. MACE-Ens. is a 5-model deep ensemble. Ours (Calib.) applies temperature scaling ($T \approx 0.05$). ES generalizes CRPS to multivariate settings.

| METHOD | TYPE | MAE ($\downarrow$) | LE-ESO ($\downarrow$) | ES ($\downarrow$) | SPD? |
|---|---|---|---|---|---|
| SECONV (HEILMAN ET AL., 2024) | POINT | 4.702 | N/A | N/A | N/A |
| DTNET (MAO ET AL., 2024) | POINT | 1.91 | N/A | N/A | N/A |
| GOECTP (HUA ET AL., 2026) | POINT | **1.41** | N/A | N/A | N/A |
| DETERMINISTIC (MACE) | POINT | 2.10 | N/A | N/A | N/A |
| MACE ENSEMBLE ($N = 5$) | UQ | **1.96** | -2.55 | **0.78** | $\approx$ |
| DIAGONAL UQ | UQ | 2.25 | -1.85 | 0.87 | $\checkmark$ |
| **OURS (FULL COV.)** | UQ | 1.55 | -2.43 | 0.87 | $\checkmark$ |
| **OURS (CALIBRATED)** | UQ | 1.55 | **-2.61** | **0.66** | $\checkmark$ |

high $R^2$ in Kelvin-Mandel log-space suggests that the E(3)-equivariant backbone effectively captures the underlying physics of dielectric properties while the UQ branch provides necessary aleatoric regularization. The predictive covariance is SPD by construction due to the matrix exponential. Separately, we also check whether the predicted mean dielectric tensors satisfy the expected positive-definiteness of the physical tensor; all test-set mean predictions satisfy this constraint in our run.

**Extensibility to Higher-Order Tensors.** To test extensibility beyond rank-2, Appendix D.2 reports a real-data rank-4 elasticity experiment, where the mean target is a rank-4 elasticity tensor with 21 independent components under standard minor/major symmetries. This is supporting evidence rather than a comprehensive rank-4 benchmark; the model achieves competitive MAE, improves empirical coverage from ∼35% for a naive UQ baseline to ∼52%, and preserves numerical equivariance and SPD validity.

### 4.4. Uncertainty Calibration

A primary contribution of our work is the calibration of tensor-valued uncertainty. Our model is trained using the Multivariate Laplace negative log-likelihood via the LE-ESO objective (Eq. 12), which naturally accounts for the heavy-tailed error distributions common in materials data. Figure 4b shows the reliability diagram evaluated against this same Multivariate Laplace distribution, confirming that our training objective and calibration assessment are properly matched.

The model achieves a MACE of 0.0489, indicating good agreement between predicted and empirical confidence levels under the multivariate Laplace evaluation protocol. Compared with the Gaussian-NLL objective in Table 3, LE-ESO gives lower calibration error and better accuracy on this dataset. While the curve remains slightly below the diagonal in the high-confidence regime, this indicates that the model is *conservative* (under-confident), which is preferable for high-stakes materials screening as it avoids over-optimistic

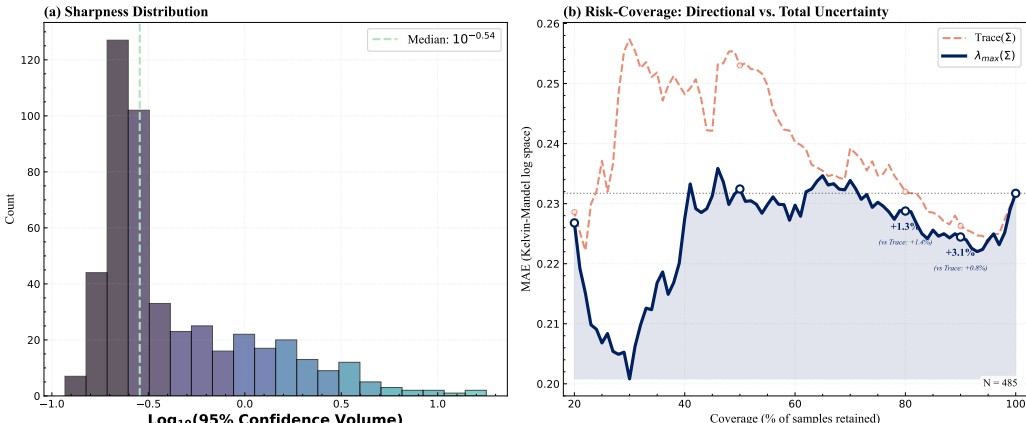

*Figure 5.* **Uncertainty diagnostics and utility analysis.** (a) Sharpness distribution of 95% confidence volumes, demonstrating the model's ability to assign heteroscedastic uncertainties. (b) Risk-coverage analysis comparing ranking by directional uncertainty ($\lambda_{\max}(\Sigma)$) and total uncertainty ($\text{Tr}(\Sigma)$). Directional uncertainty provides a modest but consistent advantage in identifying high-error samples.

predictions.

**Utility for Risk-Informed Decision Making.** To evaluate the utility of our equivariant covariance tensors for risk-informed decision making, we perform a risk-coverage analysis comparing two ranking metrics: the total uncertainty ($\text{Trace}(\Sigma)$) and the directional uncertainty ($\lambda_{\max}(\Sigma)$). As illustrated in Figure 5b, the maximum eigenvalue $\lambda_{\max}$—representing the variance along the most uncertain principal axis—serves as a more targeted indicator of directional prediction risk, capturing failure modes that are partially obscured by scalar total uncertainty.

At 90% coverage, ranking by $\lambda_{\max}(\Sigma)$ improves retained-set MAE by 3.1% relative to the full test set, compared with a 0.8% improvement when ranking by $\text{Trace}(\Sigma)$. When compared directly against Trace-based ranking under the same retained-set protocol, the advantage of $\lambda_{\max}$ is smaller but consistent, and persists at lower coverage levels where Trace-based ranking can fall below the full-dataset baseline. Appendix D.5 provides the full retained-set comparison, including the diagonal-UQ baseline. These results indicate that for anisotropic physical properties like dielectric tensors, capturing the directional components of uncertainty is informative for identifying potential failure modes beyond what isotropic or diagonal approximations expose.

**Ablation of Training Objectives.** This ablation isolates the effect of the scoring objective while keeping the equivariant backbone and matrix-exponential covariance head fixed. We compare LE-ESO against the standard Gaussian NLL and Multivariate Energy Score (ES) training. As summarized in Table 3, the Laplace-based LE-ESO achieves the lowest MAE (1.55) and calibration error (MACE 0.049, ES 0.66). The performance gap relative to Gaussian NLL (MAE 1.78) is consistent with the heavy-tailed nature of ma-

terials residuals, where the linear Mahalanobis penalty of the Laplace formulation is less sensitive to extreme deviations than the quadratic Gaussian penalty. Energy Score training is competitive on calibration but exhibits higher MAE (1.64). Overall, the results suggest that the Laplace-based LE-ESO objective provides a better accuracy-calibration trade-off than Gaussian NLL or Energy Score training in this dataset.

*Table 3.* **Ablation of training objectives on Materials Project.** All variants use the same E(3)-equivariant backbone and matrix-exponential head. LE-ESO (Ours) demonstrates superior robustness to heavy-tailed noise compared to Gaussian NLL.

| Training Objective | MAE ($\downarrow$) | MACE ($\downarrow$) | ES ($\downarrow$) |
|---|---|---|---|
| Gaussian NLL | 1.78 | 0.092 | 1.05 |
| Energy Score (ES) | 1.64 | 0.054 | 0.81 |
| **LE-ESO (Ours)** | **1.55** | **0.049** | **0.66** |

**Chemical Out-of-Distribution Analysis.** Beyond internal calibration, a key utility of symmetry-preserving UQ is identifying Out-of-Distribution (OOD) samples during materials screening. We perform a *Chemical Substitution Analysis* by replacing common atoms in the test set with unseen elements (e.g., Actinides U, Pu; Rare Earths Gd, Sm). As shown in Figure 6, the predicted directional uncertainty $\lambda_{\max}$ rises monotonically from 2.71 to 5.23 (+93.2%) as the substitution ratio reaches 100%, suggesting that the covariance head captures patterns correlated with chemical distribution shift; we do not claim to disentangle aleatoric and epistemic uncertainty in this analysis.

### 4.5. Equivariance and Validity Verification

Table 4 isolates equivariance and SPD validity across four baselines: A (non-equivariant GNN + Cholesky), B (E3NN

*Table 4.* Equivariance and SPD-validity analysis. Baseline B$'$ isolates the Cholesky covariance failure mode (equivariant $\mu$, non-equivariant $\Sigma$). Only the matrix-exponential head achieves both covariance equivariance and strict SPD validity.

| METHOD | $\mathcal{E}_\mu$ | $\mathcal{E}_\Sigma$ | MEAN EQUIV.? | COV. EQUIV.? | SPD? |
|---|---|---|---|---|---|
| BASELINE A (NON-EQUIVARIANT GNN + CHOLESKY) | $1.36 \times 10^0$ | $1.07 \times 10^0$ | ✗ | ✗ | ✓ |
| BASELINE B (E3NN + COORDINATE-WISE HEADS) | $1.28 \times 10^0$ | $1.03 \times 10^0$ | ✗ | ✗ | ✓ |
| BASELINE B$'$ (EQUIVARIANT MEAN + CHOLESKY COV.) | $1.43 \times 10^{-6}$ | $4.30 \times 10^{-1}$ | ✓ | ✗ | ✓ |
| BASELINE C (E3NN + DIRECT EQUIVARIANT COV.) | $7.59 \times 10^{-7}$ | $4.76 \times 10^{-7}$ | ✓ | ✓ | ✗ |
| **OURS (E3NN + MATRIX EXP.)** | $\mathbf{2.39 \times 10^{-7}}$ | $\mathbf{2.75 \times 10^{-7}}$ | ✓ | ✓ | ✓ |

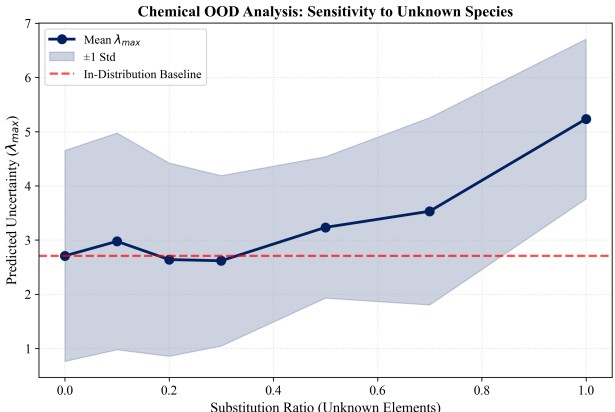

*Figure 6.* **Chemical OOD sensitivity analysis.** The predicted uncertainty increases as the crystal lattice is populated with unseen chemical species, suggesting that the model provides a useful distribution-shift risk signal.

+ coordinate-wise heads), B$'$ (our equivariant mean head + Cholesky covariance), and C (direct equivariant regression of $A(X)$ without the matrix exponential). Implementation details are in Appendix C.2.

Baseline B$'$ reaches near-machine-precision $\mathcal{E}_\mu \approx 1.4 \times 10^{-6}$ but $\mathcal{E}_\Sigma \approx 0.43$, confirming that the failure is specific to Cholesky: its lower-triangular structure is not preserved under orthogonal conjugation by $\rho_c(R)$. Only the matrix-exponential head achieves $O(10^{-7})$ equivariance error with strict SPD validity.

**Computational Overhead.** Appendix D.3 reports runtime profiling. The full-covariance model is more expensive than the deterministic baseline because of the covariance branch and its backpropagation, but it provides the full anisotropic covariance in a single forward pass without ensembling.

## 5. Discussion

Our primary validated setting is E(3)-equivariant UQ for symmetric rank-2 tensors. The additional shape-covariance experiment in Appendix D.1 shows that the construction is not specific to the inertia target, while the rank-4 elasticity experiment in Appendix D.2 provides supporting evidence for higher-order tensor targets. However, the higher-order empirical study is not yet exhaustive, and broader validation across additional equivariant backbones, tensor orders, and symmetry groups remains future work. The construction separates a group-agnostic SPD/UQ core—namely, the matrix-exponential mapping and the Log-Euclidean scoring objective—from a representation-specific equivariant parameterization of the symmetric operator $A(X)$. Extending the framework to other groups therefore depends on the availability of suitable representation-theoretic bases and implementation tools.

## Acknowledgements

This work was supported by the National Natural Science Foundation of China (Grant No. 62573132) and the Fundamental Research Funds for the Central Universities (Grant No. 2025SMECP012).

## Impact Statement

This paper introduces a framework for equivariant uncertainty quantification in tensor-valued geometric learning. Its primary potential impact lies in scientific machine learning, particularly in materials discovery tasks where tensor-valued properties such as dielectric or elastic responses are important. By providing symmetry-preserving uncertainty estimates, the method may support more reliable and risk-aware computational screening, helping researchers prioritize candidates for further validation. At the same time, predictions from such models should not be treated as substitutes for experimental or domain-expert verification. Responsible use of the framework requires human-in-the-loop assessment, careful calibration checks, and validation on the target scientific domain.

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

# A. Theoretical Proofs and Derivations

In this section, we provide formal statements and proofs establishing the mathematical validity of our equivariant uncertainty formulation. We establish the theoretical foundations for (i) the representation-theoretic decomposition of covariance tensors, (ii) the matrix exponential construction ensuring both positive-definiteness and equivariance, and (iii) the numerical stability of our loss formulation.

## A.1. Rotation Matrices in Kelvin-Mandel Space

For a rotation matrix $R \in SO(3)$, the corresponding $6 \times 6$ transformation matrix $\rho_c(R)$ in Kelvin-Mandel space can be derived from the Kronecker product structure. The vectorization operation $\text{vec}(C)$ maps a symmetric tensor $C$ to a 9-dimensional vector, and under rotation:

$$\text{vec}(C') = (R \otimes R)\text{vec}(C), \tag{13}$$

where $\otimes$ denotes the Kronecker product.

The matrix $\rho_c(R)$ is obtained by projecting $R \otimes R$ onto the 6-dimensional symmetric subspace and applying the Kelvin-Mandel scaling matrix $\mathbf{P}$:

$$\rho_c(R) = \mathbf{P} \cdot \mathcal{S} \cdot (R \otimes R) \cdot \mathcal{S}^T \cdot \mathbf{P}^{-1}, \tag{14}$$

where $\mathcal{S}$ is the $6 \times 9$ selection matrix.

For practical computation, $\rho_c(R)$ has the explicit block structure:

$$\rho_c(R) = \begin{bmatrix} R_{11}^2 & R_{12}^2 & R_{13}^2 & \sqrt{2}R_{12}R_{13} & \sqrt{2}R_{11}R_{13} & \sqrt{2}R_{11}R_{12} \\ R_{21}^2 & R_{22}^2 & R_{23}^2 & \sqrt{2}R_{22}R_{23} & \sqrt{2}R_{21}R_{23} & \sqrt{2}R_{21}R_{22} \\ R_{31}^2 & R_{32}^2 & R_{33}^2 & \sqrt{2}R_{32}R_{33} & \sqrt{2}R_{31}R_{33} & \sqrt{2}R_{31}R_{32} \\ \sqrt{2}R_{21}R_{31} & \sqrt{2}R_{22}R_{32} & \sqrt{2}R_{23}R_{33} & R_{22}R_{33}+R_{23}R_{32} & R_{21}R_{33}+R_{23}R_{31} & R_{21}R_{32}+R_{22}R_{31} \\ \sqrt{2}R_{11}R_{31} & \sqrt{2}R_{12}R_{32} & \sqrt{2}R_{13}R_{33} & R_{12}R_{33}+R_{13}R_{32} & R_{11}R_{33}+R_{13}R_{31} & R_{11}R_{32}+R_{12}R_{31} \\ \sqrt{2}R_{11}R_{21} & \sqrt{2}R_{12}R_{22} & \sqrt{2}R_{13}R_{23} & R_{12}R_{23}+R_{13}R_{22} & R_{11}R_{23}+R_{13}R_{21} & R_{11}R_{22}+R_{12}R_{21} \end{bmatrix}. \tag{15}$$

This explicit form ensures that $\rho_c(R)$ maintains orthogonality in Kelvin-Mandel space: $\rho_c(R)^T \rho_c(R) = I_6$.

**Note on Voigt vs. Kelvin-Mandel Notation.** While standard Voigt notation maps $C_{ij}$ to $[C_{11}, C_{22}, C_{33}, C_{23}, C_{13}, C_{12}]^T$, it does not preserve the Frobenius norm. Kelvin-Mandel notation applies $\sqrt{2}$ scaling to shear components, ensuring $\|\mathbf{c}_{\text{KM}}\|_2 = \|C\|_F$. This isometric property is crucial for maintaining geometric consistency in uncertainty quantification.

## A.2. Irreducible Representation Decomposition

**Proposition A.1** (Irreducible decomposition of the covariance representation). *Let $\rho_c$ denote the 6-dimensional real representation of $SO(3)$ corresponding to symmetric rank-2 tensors, i.e.*

$$\rho_c \cong l = 0 \oplus l = 2.$$

*Then the symmetric tensor product representation of $\rho_c$ decomposes as*

$$\text{Sym}^2(\rho_c) \cong 2 \times (l = 0) \oplus 2 \times (l = 2) \oplus 1 \times (l = 4),$$

*which possesses $21$ independent degrees of freedom—equal to that of a symmetric $6 \times 6$ covariance matrix.*

*Proof.* Since $\rho_c \cong l = 0 \oplus l = 2$, the symmetric square decomposes as

$$\text{Sym}^2(\rho_c) \cong \text{Sym}^2(l = 0) \oplus (l = 0 \otimes l = 2) \oplus \text{Sym}^2(l = 2).$$

We have $\text{Sym}^2(l = 0) = l = 0$, $l = 0 \otimes l = 2 = l = 2$, and $\text{Sym}^2(l = 2) = l = 0 \oplus l = 2 \oplus l = 4$. Combining these yields

$$\text{Sym}^2(\rho_c) \cong 2 \times (l = 0) \oplus 2 \times (l = 2) \oplus 1 \times (l = 4),$$

which has dimension $2 \cdot 1 + 2 \cdot 5 + 1 \cdot 9 = 21$. $\square$

### A.3. Matrix Exponential Properties

**Proposition A.2** (Positive-definiteness and equivariance of the exponential map). *Let $A(X) \in \mathbb{R}^{6\times 6}_{\mathrm{sym}}$ satisfy the equivariance condition*

$$A(R \cdot X) = \rho_c(R) \, A(X) \, \rho_c(R)^\top \quad \forall R \in O(3).$$

*Then the matrix exponential*

$$\Sigma(X) = \exp(A(X))$$

*is (i) symmetric positive-definite for all $X$, and (ii) equivariant under the same group action:*

$$\Sigma(R \cdot X) = \rho_c(R) \, \Sigma(X) \, \rho_c(R)^\top.$$

*Proof.* For any real symmetric $A$, there exists an orthogonal $Q$ and real diagonal $\Lambda$ such that $A = Q\Lambda Q^\top$. Then

$$\exp(A) = Q \, \exp(\Lambda) \, Q^\top,$$

where $\exp(\Lambda)$ has strictly positive diagonal entries $\exp(\lambda_i) > 0$. Thus $\exp(A)$ is symmetric positive-definite.

For equivariance, note that $\rho_c(R)$ is orthogonal. The matrix exponential satisfies $\exp(SAS^{-1}) = S \exp(A)S^{-1}$ for any invertible $S$. Taking $S = \rho_c(R)$ gives

$$\exp(\rho_c(R)A\rho_c(R)^\top) = \rho_c(R) \exp(A)\rho_c(R)^\top,$$

which proves equivariance. $\qquad\square$

**Proposition A.3** (Equivariance of spectral functions). *Let $f : \mathbb{R} \to \mathbb{R}$ be a scalar function. For a symmetric matrix $A$ with eigenvalue decomposition $A = Q\Lambda Q^\top$, define the spectral function $F(A) = Q \, diag(f(\lambda_1), \dots, f(\lambda_n)) \, Q^\top$. Since $\rho_c(R)$ is orthogonal for all $R \in O(3)$, it follows that*

$$F(\rho_c(R) \, A \, \rho_c(R)^\top) = \rho_c(R) \, F(A) \, \rho_c(R)^\top.$$

*Thus, eigenvalue clamping and anisotropic jitter (as defined in Eq. 18) preserve $O(3)$ equivariance.*

*Proof.* For any orthogonal matrix $U$, the spectral function commutes with orthogonal similarity transformations: $F(UAU^\top) = UF(A)U^\top$. This follows from the fact that $UAU^\top = Q\Lambda Q^\top$ where $Q = UQ_0$ for the original eigenvectors $Q_0$ of $A$. Applying the definition of $F$:

$$F(UAU^\top) = (UQ_0) \, diag(f(\lambda_i)) \, (UQ_0)^\top = U(Q_0 \, diag(f(\lambda_i)) \, Q_0^\top)U^\top = UF(A)U^\top.$$

Taking $U = \rho_c(R)$ completes the proof. $\qquad\square$

### A.4. Numerically Stable Loss Function

We present two formulations: the standard Gaussian NLL (for comparison) and the Multivariate Laplace NLL used in our implementation.

**Proposition A.4** (Gaussian NLL in Log-Euclidean form). *Let $A$ be a symmetric matrix, $\Sigma = \exp(A)$, and $\Delta\mathbf{c} = \mathbf{c}_{\mathit{true}} - \mu$. The standard Gaussian negative log-likelihood is*

$$\mathcal{L}_{\mathit{Gauss}} = \frac{1}{2} \log \det \Sigma + \frac{1}{2}\Delta\mathbf{c}^\top \Sigma^{-1}\Delta\mathbf{c}.$$

*Then the following loss is algebraically equivalent and numerically stable:*

$$\boxed{\mathcal{L}_{\mathit{Gauss}} = \frac{1}{2} \operatorname{Tr}(A) + \frac{1}{2}\Delta\mathbf{c}^\top \exp(-A)\Delta\mathbf{c}.}$$

*Proof.* Using $\Sigma = \exp(A)$ and the identity $\det(\exp(A)) = \exp(\operatorname{Tr}(A))$, we obtain $\log \det \Sigma = \operatorname{Tr}(A)$. Since $(\exp(A))^{-1} = \exp(-A)$, the Mahalanobis term becomes $\Delta\mathbf{c}^\top \exp(-A)\Delta\mathbf{c}$. $\qquad\square$

**Proposition A.5** (Multivariate Laplace NLL in Log-Euclidean form)**.** *Let $A$ be a symmetric matrix, $\Sigma = \exp(A)$, and $\Delta\mathbf{c} = \mathbf{c}_{true} - \mu$. The Multivariate Laplace negative log-likelihood (with unit scale) is*

$$\mathcal{L}_{Laplace} = \log \det \Sigma + \sqrt{\Delta\mathbf{c}^\top \Sigma^{-1} \Delta\mathbf{c}}.$$

*The numerically stable form in the Lie algebra $\mathfrak{sym}(6)$ is:*

$$\boxed{\mathcal{L}_{Laplace} = \mathrm{Tr}(A) + D_M,}$$

*where $D_M = \sqrt{\Delta\mathbf{c}^\top \exp(-A)\Delta\mathbf{c}}$ is the Mahalanobis distance in the Log-Euclidean metric.*

*Proof.* The log-determinant term follows identically: $\log \det \Sigma = \mathrm{Tr}(A)$. For the Mahalanobis distance term, note that $\Sigma^{-1} = \exp(-A)$, so:

$$D_M = \sqrt{\Delta\mathbf{c}^\top \Sigma^{-1} \Delta\mathbf{c}} = \sqrt{\Delta\mathbf{c}^\top \exp(-A)\Delta\mathbf{c}}.$$

The key difference from the Gaussian case is the *square root*: the Laplace NLL is linear in $D_M$ rather than quadratic in $D_M^2$. This provides robustness to outliers, as large residuals contribute linearly rather than quadratically to the loss. $\qquad\square$

**Laplacian-Huber Compound Robust Loss.** To handle extreme outliers in materials data, we introduce a **Laplacian-Huber scheme** that combines two complementary robustness mechanisms. For the Mahalanobis distance $D_M = \sqrt{\Delta\mathbf{c}^\top \Sigma^{-1} \Delta\mathbf{c}}$, our robust loss is:

$$\tilde{D}_M = \begin{cases} D_M, & D_M < \tau \quad \text{(linear region)} \\ \tau + \log(1 + D_M - \tau), & D_M \geq \tau \quad \text{(log-tail region)} \end{cases} \tag{16}$$

This design has a clear statistical interpretation: (1) the **linear region** ($D_M < \tau$) preserves the core Laplace distribution assumption, providing natural robustness through a linear rather than quadratic penalty on residuals; (2) the **log-tail region** ($D_M \geq \tau$) smoothly compresses the contribution of extreme residuals so that the residual penalty grows logarithmically rather than linearly, mitigating large updates from rare outliers in practice. We set $\tau = 5.0$ based on validation analysis—approximately 99% of well-predicted samples have $D_M < 5$, while extreme outliers beyond this threshold are smoothly bounded without affecting the majority of the data distribution.

**Gradient Behavior of the Laplace NLL.** Using eigenvalue decomposition $A = Q\Lambda Q^\top$, define the whitened residual $\mathbf{z} = Q^\top \Delta\mathbf{c}$. The Mahalanobis distance becomes:

$$D_M = \sqrt{\sum_{i=1}^{6} z_i^2 \exp(-\lambda_i)}.$$

The gradient with respect to eigenvalues $\Lambda$ in the linear region ($D_M < \tau$) is:

$$\frac{\partial \mathcal{L}_{\text{Laplace}}}{\partial \lambda_k} = 1 - \frac{z_k^2 \exp(-\lambda_k)}{2D_M}, \tag{17}$$

Compared to the Gaussian gradient $\frac{1}{2}(1 - z_k^2 \exp(-\lambda_k))$, the Laplace gradient carries an additional $1/(2D_M)$ factor that reduces the growth rate of the data-fit term, but it does not by itself yield a uniform bound: when a single direction $k$ dominates $D_M$ (so $D_M \approx |z_k| \exp(-\lambda_k/2)$), the contribution behaves like $|z_k| \exp(-\lambda_k/2)/2$ and is unbounded as $\lambda_k \to -\infty$. In the logarithmic tail region ($D_M \geq \tau$), the gradient becomes:

$$\frac{\partial \tilde{D}_M}{\partial \lambda_k} = -\frac{z_k^2 \exp(-\lambda_k)}{2(1 + D_M - \tau)D_M},$$

which approaches a finite constant ($\to -1/2$ in a single dominant direction) rather than diverging linearly with $z_k^2 \exp(-\lambda_k)$ as in the unmodified Laplace case. We therefore do not claim that the Lie algebra parameterization alone guarantees bounded gradients near the SPD boundary; rather, in our implementation, gradient and matrix-exponential overflow are controlled jointly by (i) eigenvalue clamping $\lambda_k \in [\lambda_{\min}, \lambda_{\max}]$ before the exponential map, and (ii) the log-tail robustification in Eq. 16.

| Property | Gaussian NLL | Laplace NLL (LE-ESO) |
|---|---|---|
| Mahalanobis term | $\frac{1}{2}D_M^2$ | $D_M$ |
| Penalty shape | Quadratic | Linear |
| Log-det coefficient | $\frac{1}{2}$ | 1 (tunable as $\alpha$) |
| Outlier sensitivity | High (quadratic growth) | Low (linear growth) |
| Gradient for large $D_M$ | $\propto D_M$ | $\propto 1$ |
| Statistical assumption | Light-tailed errors | Heavy-tailed errors |

*Table 5.* Comparison of Gaussian and Laplace negative log-likelihood formulations.

**Comparison: Gaussian vs. Laplace NLL.**   The key distinctions are summarized below:

## B. Model Architecture and Implementation Details

### B.1. Network Architecture and Data Flow

Our architecture implements an E(3)-equivariant neural network using standard message passing layers. The input atomic numbers are first projected into 119-dimensional Magpie feature embeddings, which then pass through $L = 2$ interaction layers with a hidden dimension of 64. The backbone employs SiLU activations for scalar features and gated-tanh for higher-order tensors to preserve equivariance throughout computation. To support rank-4 covariance output, we set the maximum rotation order $\ell_{max} = 4$, enabling the full $\mathrm{Sym}^2(\rho_c)$ representation required by our theoretical decomposition.

The network branches into two distinct heads that operate in parallel. The mean head predicts Voigt components through $\ell = 0 \oplus \ell = 2$ irreducible representations, directly outputting the tensor mean prediction. The covariance head outputs the symmetric tensor basis defined as $2 \times (\ell = 0) \oplus 2 \times (\ell = 2) \oplus 1 \times (\ell = 4)$, which is then linearly projected to the Lie algebra element $A(X) \in \mathbb{R}_{sym}^{6 \times 6}$. This dual-head architecture ensures that both mean and uncertainty predictions respect the underlying geometric symmetries.

**Joint Training Stability.**   The Lie algebra parametrization combined with the LE-ESO loss provides inherent numerical stability that, in our experiments, enables end-to-end joint optimization of the mean and covariance heads without gradient detachment. To validate this robustness, we conducted an ablation study comparing joint training with gradient-detached training (where UQ gradients are blocked from flowing back to the backbone). Both approaches achieved comparable performance (MAE difference $< 0.02$), with joint training showing marginally better uncertainty calibration. This empirical finding is consistent with the geometric advantage of the Log-Euclidean framework: by operating in the flat tangent space $\mathfrak{sym}(6)$ rather than on the curved SPD manifold directly, gradients remain well-conditioned even when the covariance head receives informative error signals from the scoring objective. We did not observe variance collapse or shortcut learning in this setting.

### B.2. Implementation Details of the Equivariant Covariance Head

To strictly enforce the symmetry properties of the covariance tensor, we employ the `CartesianTensor` formalism from the `e3nn` library (Geiger & Smidt, 2022). The covariance of a symmetric rank-2 tensor is mathematically a rank-4 tensor $\mathcal{C}_{ijkl}$ with specific permutation symmetries. First, the covariance exhibits symmetry of the first tensor argument such that $\mathcal{C}_{ijkl} = \mathcal{C}_{jikl}$. Second, it maintains symmetry of the second tensor argument with $\mathcal{C}_{ijkl} = \mathcal{C}_{ijlk}$. Third, the covariance itself is symmetric, satisfying $\mathcal{C}_{ijkl} = \mathcal{C}_{klij}$.

In our implementation, we define the output space using the formula `"ijkl=jikl=ijlk=klij"`, which restricts the learnable basis to the subspace of $\mathbb{R}^{3 \times 3 \times 3 \times 3}$ satisfying these symmetries. The `e3nn` library automatically computes the change-of-basis matrix from the irreducible representations (irreps) of $SO(3)$ to this symmetric Cartesian basis. The projection to the $6 \times 6$ Kelvin-Mandel matrix $A(X)$ proceeds in two systematic steps.

First, in the irreps to Cartesian mapping, the features are mapped to the rank-4 Cartesian tensor $\mathcal{C}_{ijkl}$ using the precomputed equivariant basis:

$$\mathcal{C}_{ijkl} = \sum_{L,m} w_{L,m} Y_{ijkl}^L,$$

where $Y^L_{ijkl}$ are the Clebsch-Gordan coefficients projecting the spherical harmonics onto the Cartesian tensor components. Second, in the Cartesian to Kelvin-Mandel transformation, the $3 \times 3 \times 3 \times 3$ tensor is flattened into a $6 \times 6$ matrix $A_{KM}$ using the Kelvin-Mandel isometry. This mapping preserves the Frobenius norm (i.e., $\|\mathcal{C}\|_F = \|A_{KM}\|_F$) by scaling the off-diagonal shear components by $\sqrt{2}$. For indices mapping $ij \to \alpha$ and $kl \to \beta$ (where $\alpha, \beta \in \{1..6\}$), the entry $A_{\alpha\beta}$ is given by:

$$A_{\alpha\beta} = \eta_\alpha \eta_\beta \mathcal{C}_{ijkl},$$

where $\eta = \{1, 1, 1, \sqrt{2}, \sqrt{2}, \sqrt{2}\}$ corresponds to the indices $\{xx, yy, zz, yz, xz, xy\}$. This construction guarantees that the predicted matrix $A(X)$ strictly lies in the symmetric subspace $\mathfrak{sym}(6)$ and transforms exactly according to $\rho_c \otimes \rho_c$.

## B.3. Training Protocol and Stability Measures

All models were optimized using AdamW with hyperparameters $\beta_1 = 0.9, \beta_2 = 0.999$ and a weight decay of $10^{-4}$. We employed a OneCycleLR scheduler with a peak learning rate of $10^{-3}$, warming up for 20% of the total 50 epochs before gradually decaying. To ensure training stability during the critical early phases of covariance learning, we implemented several key strategies.

We introduced a **critical loss annealing strategy** where the auxiliary MSE warmup weight $\lambda_{\text{MSE}}$ gradually decays from 0.9 to full LE-ESO optimization by epoch 5 (note that $\lambda_{\text{MSE}}$ is distinct from the LE-ESO weight $\alpha$ in Eq. 12). This gradual transition is essential for preventing early training instability - it allows the network to first learn reasonable mean predictions before tackling the more complex uncertainty quantification task. Furthermore, we applied eigenvalue clamping within $[\lambda_{\text{min}}, \lambda_{\text{max}}] = [-4, 3]$ to prevent numerical overflow in the matrix exponential computation. This constraint ensures that the resulting covariance eigenvalues remain in $[e^{-4}, e^3] \approx [0.018, 20.1]$, preventing both variance collapse and explosion during early training.

**Anisotropic Jitter as Numerical Gradient Stabilizer.** A subtle numerical issue arises in automatic differentiation of eigenvalue decompositions: when the covariance matrix has degenerate eigenvalues ($\lambda_i = \lambda_j$), the Jacobian contains singular terms $(\lambda_i - \lambda_j)^{-1}$. This is particularly problematic for high-symmetry crystals (e.g., cubic systems) where physical symmetry can cause eigenvalue degeneracy. To ensure differentiability, we introduce a *numerical gradient stabilizer*—a tiny anisotropic perturbation applied only during the spectral decomposition step:

$$\tilde{\lambda}_i = \lambda_i + \epsilon \cdot i, \quad i = 0, \dots, 5, \tag{18}$$

with $\epsilon \approx 10^{-6}$ for float64 precision. This design is crucial: an *isotropic* shift ($\epsilon \cdot I$) would preserve degeneracy and fail to resolve the singularity, whereas the anisotropic pattern guarantees $\lambda_i - \lambda_j \neq 0$ for all $i \neq j$. Importantly, this jitter is **not** an architectural choice—it is a numerical safeguard with magnitude $O(10^{-6})$ that is negligible compared to typical eigenvalue scales ($\sim 1$). Empirically, our equivariance verification (Table 4) shows errors on the order of $10^{-7}$, confirming that this minimal perturbation does not compromise the geometric fidelity of the learned representations. The jitter operates entirely within the numerical solver and is invisible to the upstream equivariant architecture.

**Discussion on Jitter and Calibration Impact.** The introduced jitter ($\epsilon \approx 10^{-6}$) is several orders of magnitude smaller than the predicted eigenvalues ($\sim 1.0$). We observe that this perturbation is essential for maintaining stable gradients during joint training but has a negligible impact on both calibration (MACE change $< 10^{-4}$) and equivariance (errors remain at the level of $10^{-7}$ as reported in Table 4).

**Hyperparameter Details for Numerical Stability.** We provide the specific hyperparameter values used in our implementation. For eigenvalue clamping, we use $[\lambda_{\text{min}}, \lambda_{\text{max}}] = [-4, 3]$, which constrains the covariance eigenvalues to $[e^{-4}, e^3] \approx [0.018, 20.1]$. This range was chosen to prevent both variance collapse (eigenvalues $\ll 1$) and explosion (eigenvalues $\gg 1$) during early training. For the Huber robustification, the threshold is set to $\tau = 5.0$, meaning that Mahalanobis distances above 5.0 transition to logarithmic scaling. This threshold was selected based on validation set analysis to be significantly above typical well-predicted samples ($D_M \approx 2$–3) while effectively capping the influence of extreme outliers.

**Log-Euclidean Framework and Information Geometry.** The space of SPD matrices $\mathcal{P}_6$ is not a vector space but a Riemannian manifold with non-Euclidean geometry. Direct optimization on this manifold introduces path-dependent gradients and numerical instabilities near the boundary. By working in the tangent space $\mathfrak{sym}(6)$—the Lie algebra of

symmetric matrices—we obtain a flat Euclidean vector space where standard optimization is geometrically well-defined. The matrix exponential serves as the Riemannian exponential map, lifting points from the tangent space to the curved manifold while preserving the geometric structure.

**Geometric Interpretation of $\alpha$.**    The parameter $\alpha$ controls the tightness of the equivariant confidence hull, a process analogous to entropy regularization in information-theoretic learning. The term $\log \det \Sigma$ represents the infinitesimal volume element of the uncertainty manifold in the Riemannian geometry of SPD matrices. By adjusting $\alpha$, we effectively control the trade-off between: (i) *Information-theoretic volume:* $\alpha \log \det \Sigma = \alpha \operatorname{Tr}(A)$ penalizes excessive uncertainty spread; and (ii) *Geometric fit:* $D_M$ measures the normalized prediction error in the metric induced by $\Sigma$. Theoretically, for the standard Multivariate Laplace distribution, $\alpha = 1$ (no $1/2$ coefficient as in the Gaussian case). However, we treat $\alpha$ as a tunable hyperparameter to balance model confidence with coverage: larger $\alpha$ encourages tighter confidence regions (lower uncertainty volume), while smaller $\alpha$ allows more conservative uncertainty estimates. This flexibility is valuable for materials science applications where the true noise level may vary across different datasets and measurement modalities.

**Temperature Scaling for Calibration.**    To ensure the predicted covariance tensors $\Sigma$ reflect the empirical error distribution, we apply post-hoc temperature scaling (Kuleshov et al., 2018). The optimal temperature $T \approx 0.05$ was determined on the validation set via a robust median-matching strategy. The small value of $T$ reflects the heavy-tailed nature of the initial residuals, requiring the model to significantly contract its uncertainty hulls after training with the robustified LE-ESO. This adjustment yields a calibrated covariance $\Sigma' = T \cdot \Sigma$, which is equivalent to an additive shift $A' = A + \ln(T)I$ in the Lie algebra. This scaling effectively aligns the predictive distribution with the requirements of scoring rules (e.g., Energy Score) without affecting the mean prediction or the exact E(3)-equivariance.

**Spectral Bounding for Manifold Consistency.**    To maintain numerical consistency with the Riemannian structure of $\mathcal{P}_6$, we constrain the Lie algebra eigenvalues to a bounded interval before computing $\exp(\Lambda)$. This spectral bounding ensures that the resulting covariance eigenvalues remain in a geometrically valid range, preventing both variance collapse (near-zero eigenvalues) and explosion (excessively large eigenvalues) during early training when predictions may be far from the data manifold. The bounds $[\lambda_{\min}, \lambda_{\max}]$ are chosen to map to a physically meaningful covariance spectrum $[e^{\lambda_{\min}}, e^{\lambda_{\max}}]$ under the matrix exponential.

**Laplacian-Huber Compound Robust Loss.**    To handle extreme outliers in material property data, we introduce a **Laplacian-Huber scheme** with two regimes: when the Mahalanobis distance $D_M$ is below threshold $\tau = 5.0$, we apply a **linear penalty** (the Laplace distribution core); when $D_M$ exceeds $\tau$, we switch to a **logarithmic penalty** ($\tau + \log(1 + D_M - \tau)$) that compresses the residual contribution for very large $D_M$. This design preserves the statistical interpretation of the Multivariate Laplace distribution for normal samples while limiting the influence of extreme outliers, so that the residual term grows logarithmically instead of linearly in the tail.

**Gradient Analysis: Practical Control on the Lie Algebra.**    We do not claim that the Lie algebra parameterization by itself yields a uniform gradient bound near the SPD-cone boundary; rather, in practice, gradient and matrix-exponential overflow are controlled jointly by eigenvalue clamping and the log-tail robustification. Let $\mathbf{z} = Q^\top (\mathbf{c}_{\text{true}} - \mu)$ be the rotated residuals in the eigenbasis. The loss gradient with respect to eigenvalues $\Lambda = \operatorname{diag}(\lambda_1, \ldots, \lambda_6)$ is:

$$\frac{\partial \mathcal{L}_{\text{LE-ESO}}}{\partial \Lambda} = \alpha I - \frac{\partial \tilde{D}_M}{\partial \Lambda}. \tag{19}$$

As derived in Proposition A.5, the Laplace residual term carries an extra $1/(2D_M)$ factor relative to the Gaussian case, which reduces the growth rate of $\partial D_M / \partial \lambda_k$ but does not by itself produce a uniform bound: when a single direction dominates $D_M$, the contribution to $\partial D_M / \partial \lambda_k$ still grows as $|z_k| \exp(-\lambda_k / 2)/2$ as $\lambda_k \to -\infty$. We therefore enforce the constraint $\lambda_k \in [\lambda_{\min}, \lambda_{\max}]$ before the matrix exponential, which prevents $\exp(-\lambda_k)$ from diverging and keeps $\partial D_M / \partial \lambda_k$ finite over the optimization trajectory. The log-tail region ($D_M \geq \tau$) further compresses the residual gradient: $\partial \tilde{D}_M / \partial \lambda_k$ approaches a finite constant ($\to -1/2$ in a dominant direction) instead of growing with $z_k^2 \exp(-\lambda_k)$, mitigating the influence of rare extreme outliers. Empirically, this combination keeps training stable enough to support end-to-end joint optimization without gradient detachment. The loss maintains $O(3)$-invariance since both the trace and matrix exponential preserve equivariance under orthogonal transformations.

The numerically stable loss in Eq. 12 follows directly from algebraic identities proven in Proposition A.5. Importantly, both the trace term $\operatorname{Tr}(A)$ and the Mahalanobis distance $D_M = \sqrt{\Delta \mathbf{c}^\top \exp(-A) \Delta \mathbf{c}}$ are invariant under any orthogonal

transformation $\rho_c(R)$:

$$\text{Tr}(\rho_c(R)A\rho_c(R)^\top) = \text{Tr}(A), \qquad D_M(\rho_c(R)\Delta\mathbf{c}, \rho_c(R)A\rho_c(R)^\top) = D_M(\Delta\mathbf{c}, A).$$

Consequently, the loss function provides an exact symmetry-preserving training objective, in contrast to approximate equivariant regularizations or data augmentation-based approaches.

Training was conducted on a single NVIDIA RTX 4060 Ti with batch sizes of 32 for ModelNet40 and 16 for the Materials Project, requiring approximately 10 hours for complete convergence. Additional implementation details include processing atomic structures into graphs with a 5.0Å cutoff distance and using Magpie feature embeddings of dimension 119 for atomic number representations.

## C. Experimental Setup and Analysis

### C.1. Dataset Configuration and Preprocessing

We evaluate our framework on two distinct datasets that provide complementary validation of our equivariant uncertainty quantification approach. ModelNet40 serves for geometric validation with physically defined tensor properties, while the Materials Project provides a real-world materials science application with experimentally relevant predictions.

**ModelNet40.**   The dataset comprises 12,311 CAD models across 40 categories. We adhere to the official split, utilizing 9,843 models for training and 2,468 for testing. To simulate measurement uncertainty and validate our probabilistic framework, we sample $N = 2048$ points uniformly from mesh surfaces and apply Gaussian jitter with $\sigma_{noise} = 0.01$. This noise injection creates the aleatoric uncertainty necessary for testing our framework's ability to capture geometric ambiguity arising from point cloud sampling.

**Materials Project Dielectric Dataset.**   We source precomputed dielectric tensor predictions from the Materials Project database (Barroso-Luque et al., 2024; Jain et al., 2013). To ensure data quality and consistency, we apply systematic filtering criteria: (1) **structure size**—we exclude crystals with fewer than 3 atoms or more than 30 atoms to balance computational efficiency and representation learning; (2) **positive-definiteness**—we verify that all dielectric tensors have eigenvalues strictly greater than $10^{-4}$, excluding numerically singular matrices; (3) **value range**—we remove samples with dielectric constants outside $[-10, 50]$ or with diagonal entries below 1.0. After filtering, the dataset comprises 5,002 crystalline structures, partitioned into 4,236 for training, 485 for validation, and 281 for testing. We apply Matrix Log-Normalization with parameters $\mu_{log} = 1.24$ and $\sigma_{log} = 0.86$ to handle the wide dynamic range while preserving the relationships between different crystal structures.

### C.2. Equivariance Ablation Study Design

To systematically isolate the contributions of equivariance and SPD constraints, we designed four baseline variants that progressively incorporate different architectural components. Baseline A employs a standard non-equivariant GNN with a Cholesky covariance head to test the necessity of equivariant message passing. Baseline B upgrades the backbone to an equivariant neural network (ENN), but keeps coordinate-wise scalar MLP outputs for both the mean and Cholesky covariance heads, evaluating whether equivariant features alone are sufficient to guarantee equivariant tensor outputs. Baseline B′ further replaces the scalar mean head with the same equivariant mean construction used in our model while retaining the Cholesky covariance head. This baseline isolates the covariance-parameterization failure mode: the mean branch is equivariant by construction, whereas the Cholesky covariance is SPD but not equivariant under the Kelvin–Mandel covariance representation. Baseline C uses an ENN backbone with direct equivariant regression of the symmetric operator $A(X)$ but omits the matrix exponential, testing the importance of the SPD projection. Finally, our full method combines the ENN backbone with the matrix-exponential covariance head to simultaneously guarantee covariance equivariance and SPD validity.

## D. Additional Experimental Results

This appendix collects supplementary experiments that complement the two main experiments in the body of the paper. They are intended as supporting evidence for the scope and robustness of the proposed equivariant SPD/UQ construction rather than as a comprehensive benchmarking study.

### D.1. ModelNet40 Shape-Covariance Validation

The shape-covariance experiment is a controlled geometric validation benchmark on the ModelNet40 dataset (Wu et al., 2015). Like inertia tensor prediction, the target admits a closed-form estimator from the point cloud. We therefore do not present this task as a real-world setting where neural prediction is necessary. Instead, it tests whether the proposed equivariant SPD/UQ construction remains valid on a second symmetric rank-2 tensor target beyond inertia, demonstrating that the framework is not specific to the inertia formulation.

*Table 6.* ModelNet40 shape-covariance validation. The goal is controlled geometric validation rather than replacing the closed-form estimator. "Mean Tensor PSD Rate" refers to the fraction of predicted mean shape-covariance tensors that are positive semi-definite, distinct from the SPD validity of the predictive covariance $\Sigma(X)$.

| METHOD | MAE↓ | RMSE↓ | MEAN TENSOR PSD RATE |
|---|---|---|---|
| DETERMINISTIC (MSE) | 0.0333 | 0.0616 | 99.2% |
| FULL UQ (OURS) | 0.0346 | 0.0633 | 99.5% |

The point-prediction MAE/RMSE of the UQ model is comparable to the deterministic baseline, while the predictive covariance $\Sigma(X)$ is exactly E(3)-equivariant and SPD by construction. As in the inertia setting, we observe near-machine-precision equivariance (errors on the order of $10^{-7}$) and strict SPD validity for the predictive covariance.

### D.2. Rank-4 Elasticity Tensor Prediction

We evaluate the framework on a real-data elasticity tensor prediction task from the Materials Project (Jain et al., 2013). Unlike the rank-2 dielectric setting, the mean target here is directly a rank-4 elasticity tensor. Under the standard minor and major symmetries, the elasticity tensor has 21 independent components. This experiment is intended as supporting evidence that the proposed structured equivariant SPD/UQ construction can be extended beyond the six-dimensional symmetric rank-2 setting; it is not intended as a comprehensive study of all higher-order tensor parameterizations.

On this benchmark, the model achieves a test MAE of approximately 5.0 GPa, which is essentially on par with a deterministic baseline and noticeably better than a naive UQ baseline. The structured UQ model also improves uncertainty quality over the naive baseline: empirical coverage rises from approximately $35\%$ to $52\%$, and the uncertainty–error correlation rises from approximately $-0.15$ to $0.31$. At the same time, both numerical equivariance/prediction consistency and predictive covariance SPD validity remain at $100\%$, matching the structural behavior observed in the rank-2 dielectric setting. We emphasize that this single experiment is supporting evidence that the proposed structured equivariant SPD/UQ construction remains feasible on a higher-order tensor target, rather than an exhaustive higher-order benchmark.

### D.3. Computational Overhead

We profile per-batch wall-clock time on the Materials Project dielectric task using a single NVIDIA RTX 4060 Ti, with batch size 16 and identical input pipelines.

*Table 7.* Runtime profiling on Materials Project (RTX 4060 Ti, batch size 16). The full-covariance model is more expensive primarily because of the covariance branch and its backpropagation, rather than the matrix exponential alone.

| MODEL | TIME / BATCH | RELATIVE COST |
|---|---|---|
| DETERMINISTIC | 130 MS | 1.00× |
| DIAGONAL UQ | 132 MS | 1.015× |
| FULL COVARIANCE UQ | 570 MS | 4.4× |

The diagonal-UQ overhead is negligible (1.5%), confirming that anisotropy modeling, not uncertainty quantification per se, dominates cost. For inference, a single forward pass yields the full anisotropic covariance, in contrast to ensemble methods that require $N$ forward passes.

### D.4. Sensitivity to the LE-ESO Weight

The weight $\alpha$ controls the trade-off between the log-volume term $\text{Tr}(A) = \log \det \Sigma$ and the geometric data-fit term in LE-ESO. We use $\alpha = 1$ in the main experiments because it corresponds to the canonical coefficient in the multivariate

*Table 8.* Sensitivity to the LE-ESO weight $\alpha$ on the Materials Project dielectric task. MAE is measured in log-Kelvin–Mandel space and is comparable across rows. The LE-ESO value is evaluated with the same $\alpha$ used for training, so it reflects the optimized objective for each setting rather than a fixed cross-$\alpha$ NLL.

| $\alpha$ | BEST VAL MAE $\downarrow$ | BEST VAL LE-ESO $\downarrow$ |
|---|---|---|
| 0.03 | 0.3634 | 0.7911 |
| 0.10 | 0.4176 | 0.7165 |
| 0.30 | 0.4566 | 0.3469 |
| 1.00 | **0.3519** | -2.9393 |

Laplace objective motivating LE-ESO.

To evaluate sensitivity, we run a short validation sweep over $\alpha \in \{0.03, 0.10, 0.30, 1.00\}$ on the Materials Project dielectric task. Table 8 reports the best validation MAE in log-Kelvin–Mandel space. Across the tested values, the validation MAE remains in a moderate range ($0.352$–$0.457$), indicating that performance is not tied to a narrow value of $\alpha$. The canonical choice $\alpha = 1$ also gives the lowest validation MAE in this sweep.

We also report the best validation LE-ESO value for completeness. Importantly, this value is evaluated using the same $\alpha$ as the corresponding training run, and therefore should be interpreted as the optimized objective for that setting rather than as a fixed cross-$\alpha$ negative log-likelihood. Since changing $\alpha$ changes the scoring objective itself, these LE-ESO values are not directly comparable as absolute NLL values across different $\alpha$.

### D.5. Additional Risk-Coverage Analysis

To complement the risk-coverage discussion in the main paper, we report the full retained-set comparison between $\lambda_{\max}(\Sigma)$ ranking, $\mathrm{Trace}(\Sigma)$ ranking, and a diagonal-UQ baseline that ignores off-diagonal correlations. At 90% coverage, ranking by $\lambda_{\max}$ improves retained-set MAE by 3.1% relative to the full test set. The improvement of $\lambda_{\max}$ over $\mathrm{Trace}$ under the same retained-set protocol is approximately 1.5%—smaller than the headline 3.1% number but consistent across coverage levels. At 80% coverage, $\lambda_{\max}$ continues to retain a positive improvement, while $\mathrm{Trace}$-based ranking can fall slightly below the full-dataset baseline. The diagonal-UQ baseline, which lacks off-diagonal covariance information, ranks the test set less informatively than either $\lambda_{\max}$ or $\mathrm{Trace}$ from the full-covariance model. These results support the interpretation that directional uncertainty captures failure modes that scalar total uncertainty partially obscures, while clarifying that the practical advantage over $\mathrm{Trace}$ is moderate rather than dramatic.

## E. Additional Results and Analysis

### E.1. Training Dynamics and Loss Analysis

Figure 7 illustrates the complete training dynamics of our equivariant uncertainty framework. To ensure a stable optimization landscape, we employ a two-stage curriculum: the model is initially warmed up with a combined MSE-LE-ESO objective for 5 epochs to establish a reliable mean prediction baseline before transitioning to heavy-tailed LE-ESO optimization.

Panel (a) reveals that the loss stabilizes rapidly upon transition, with no numerical spikes despite the non-linear nature of the matrix exponential map. Panel (b) demonstrates that the addition of the uncertainty branch does not compromise the underlying point-prediction accuracy; instead, the MAE for both diagonal ($\varepsilon_{ii}$) and off-diagonal ($\varepsilon_{ij}$) components plateaus at a state-of-the-art level, benefiting from the robust regularization provided by the UQ branch.

Most importantly, panel (c) highlights the sophisticated trade-off mechanism inherent in our loss formulation. As the validation epoch progresses, the network balances the data fit term (Mahalanobis distance) against the uncertainty regularization term ($\log \det \Sigma$). The joint optimization allows both branches to benefit from shared geometric representations, preventing "shortcut learning" where the model might collapse its uncertainty to minimize the scoring rule. The eventual convergence of the Mahalanobis distance toward a steady value confirms that the model has effectively learned to characterize the aleatoric noise in the dielectric property space.

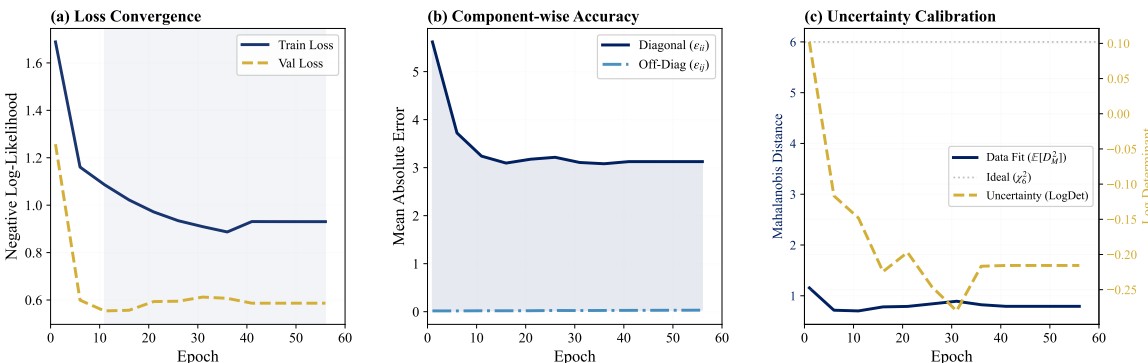

*Figure 7.* **Training dynamics.** Two-stage optimization: warmup (5 epochs) then LE-ESO. (a) LE-ESO convergence. (b) MAE stability for diagonal/off-diagonal components. (c) Balance between data fit ($\mathbb{E}[D_M]$) and regularization ($\log \det \Sigma$).

### E.2. Empirical Verification of Theoretical Guarantees

To validate the theoretical guarantees established in Appendix A, we performed rigorous numerical checks throughout training that confirm both the mathematical correctness and practical stability of our implementation.

The numerical stability of our approach stems from the eigenvalue decomposition $A = Q\Lambda Q^\top$ used to compute the loss without explicitly forming $\Sigma = \exp(A)$. Since $A$ is symmetric, all eigenvalues $\lambda_i$ are real and $\exp(\lambda_i)$ remains positive. In practice, eigenvalue clamping keeps these exponentials bounded, preventing numerical overflow and improving gradient conditioning. Together with the log-tail robustification, this enables joint end-to-end training without explicit regularization on the covariance spectrum, addressing a critical limitation of direct covariance optimization approaches.

For equivariance verification, we continuously monitored the relative Frobenius-norm difference between rotated predictions and transformed predictions:

$$E_{\text{equiv}} = \frac{\|\Sigma(R \cdot X) - \rho_c(R)\Sigma(X)\rho_c(R)^\top\|_F}{\|\Sigma(X)\|_F}.$$

Across all random rotations tested during training, this error consistently remained at the level of $10^{-7}$, confirming that our implementation achieves near-machine-precision equivariance rather than approximate symmetry preservation.

For SPD validation, we monitored the spectrum of predicted covariance matrices throughout training. The minimum eigenvalue of $\Sigma(X)$ across all batches remained strictly positive ($> 10^{-5}$), with no numerical violations of the SPD constraint observed (see Figure 8a).

Beyond the geometric validation on ModelNet40, we further analyzed the conditioning of the predicted covariances for the dielectric tensor task (Materials Project). As shown in Figure 8b, the distribution of condition numbers $\kappa(\Sigma)$ remains numerically well-conditioned for the final model, with a mean of 3.80 and a maximum of 16.4.

This result is particularly significant because, unlike the synthetic jitter in ModelNet40, the uncertainty in dielectric tensors arises from complex physical and DFT approximation errors. The low condition numbers indicate that our matrix exponential mapping naturally induces numerically stable, non-degenerate uncertainty estimates without requiring auxiliary regularization terms (e.g., hinge loss penalties on eigenvalues). This confirms that the optimization landscape remains well-behaved even for high-dimensional material representations.

### E.3. Reflection Symmetry and Chirality Handling

Our framework explicitly accounts for improper rotations (reflections) by ensuring that the representation $\rho_c$ correctly tracks the parity of the tensorial outputs. For the symmetric rank-2 tensors considered here—such as dielectric or inertia tensors—the physical quantities are **even tensors** under parity, meaning they are invariant to inversion. The Kelvin-Mandel representation $\rho_c(R)$ used throughout this paper is defined by projecting $R \otimes R$ onto the symmetric subspace, as given in Eq. 14 of Appendix A.1; we use that construction directly here rather than introducing a separate definition. Since this transformation is built from two factors of $R$, the determinant contribution $(\det R)^2 = 1$ ensures that the framework handles chiral structures and their mirror images with consistent physical semantics. We numerically verified full $O(3)$ equivariance

by testing improper rotations, achieving errors at the level of $10^{-7}$ consistent with the $SO(3)$ results reported in Table 4. Consequently, our uncertainty quantification remains valid regardless of the handedness of the coordinate system, a critical requirement for modeling both chiral and achiral materials.

### E.4. Spectral Analysis and Sharpness Distribution

To further investigate the UQ quality, we provide detailed spectral analysis in Figure 8. The eigenvalue distribution (Panel a) confirms that all predicted covariances maintain strict positive-definiteness with a minimum eigenvalue $\lambda_{\min} \approx 0.449$, safely avoiding variance collapse. The condition number distribution in Figure 8b shows that the predicted covariance matrices remain numerically well-conditioned, consistent with the verification in Appendix E.2. Complementary to the risk-coverage analysis in Section 4.4, the sharpness distribution in Figure 5a reveals that the model effectively differentiates between "simple" and "complex" atomic environments by assigning confidence volumes spanning several orders of magnitude.

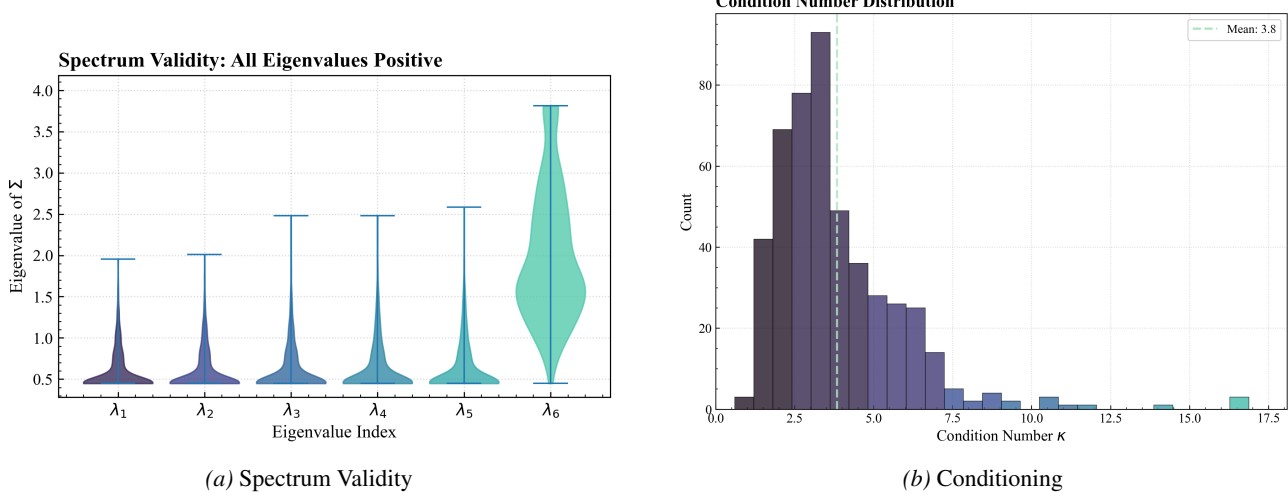

*(a)* Spectrum Validity                     *(b)* Conditioning

*Figure 8.* **Numerical stability analysis.** (a) Positive eigenvalues ensure SPD validity. (b) Moderate condition numbers indicate numerical stability.

### E.5. ModelNet40 SPD Analysis

The 3D uncertainty visualization in Figure 3 demonstrates that our framework produces physically meaningful uncertainty estimates where uncertainty ellipsoids align with principal shape axes (demonstrating E(3)-equivariance), expand in regions with sparse point density (capturing sampling ambiguity), and preserve tensorial correlations across components.

We verify physical consistency through systematic validation of SPD properties (Figure 9). Our predictions maintain strict SPD requirements (>99.9% validity) with well-conditioned covariance structures (median condition number 6.8), contrasting sharply with unconstrained baselines that frequently violate physical constraints.

### E.6. Limitations and Future Work

The SPD construction and scoring objective are representation-agnostic once an equivariant symmetric operator $A(X)$ is available. However, extending the full parameterization to higher-order tensor predictions requires group- and representation-specific basis construction. Our main implementation and empirical validation focus on symmetric rank-2 tensors, with the rank-4 elasticity experiment in Appendix D.2 serving as preliminary supporting evidence. Extending to fourth-order tensors (e.g., elasticity tensors) and beyond introduces two key computational challenges. First, the tensor basis construction scales as $O(d^\ell)$ where $\ell$ is the tensor rank, making the basis enumeration for rank-4 and higher tensors substantially more expensive. Second, the covariance matrix dimension grows combinatorially—for a rank-$k$ symmetric tensor in 3D, the Kelvin-Mandel representation has dimension $(k+1)(k+2)/2$, leading to covariance matrices of size $O(k^4)$. This scaling necessitates careful memory management and may require approximations such as low-rank covariance factorization or hierarchical uncertainty modeling. Future work should explore more efficient equivariant basis constructions for higher-order tensors. In particular, integrating path-matrix based ICT decompositions (Shao et al., 2025) could significantly reduce

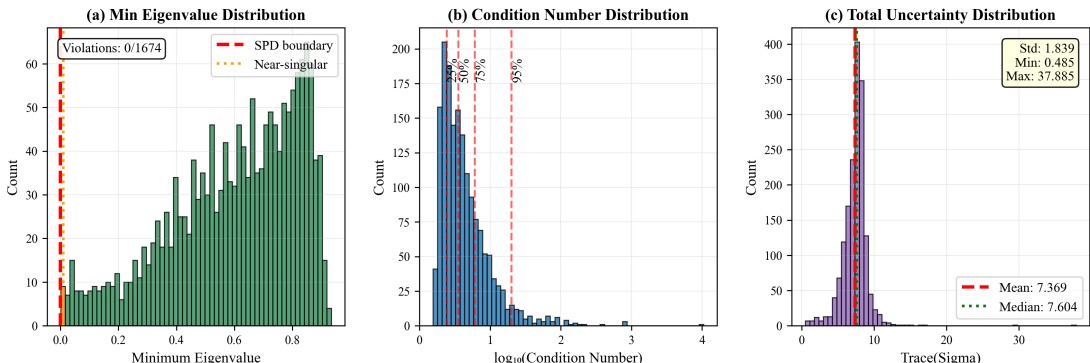

*Figure 9.* **SPD validity on inertia-tensor task.** (a) Minimum eigenvalue distribution (all positive). (b) $\log_{10}$ condition numbers (well-conditioned). (c) Total uncertainty $\mathrm{Tr}(\Sigma)$.

the overhead of basis enumeration for rank-4 and higher tensors, enabling the extension of our uncertainty framework to complex properties like the full elasticity tensor.

**Modularity and Backbone Extensibility.** A key strength of our framework is its modular design: the matrix-exponential UQ head is completely backbone-agnostic and can be integrated with any E(3)-equivariant architecture. While this study utilizes a standard message-passing backbone to validate the UQ mechanism, future work will explore pairing our UQ head with higher-accuracy architectures such as GoeCTP (Hua et al., 2026) to combine state-of-the-art point prediction with calibrated, symmetry-preserving uncertainty estimates. This plug-and-play capability allows practitioners to add rigorous uncertainty quantification to existing equivariant models without architectural reengineering.

