# OpenReview forum: "Equivariant Covariance Tensors: Guaranteed SPD Uncertainty for Tensor-Valued Geometric Learning"
_ICML.cc/2026/Conference — ICML 2026 regular_

### Official Review · Reviewer_TpCA · 2026-03-04

**Soundness:** 3
**Presentation:** 3
**Significance:** 3
**Originality:** 3
**Overall Recommendation:** 4
**Confidence:** 3

**Summary:**

This paper studies and targets uncertainty quantification for tensor-valued prediction in geometric learning settings where predictions must satisfy group equivariance. The authors propose a framework for E(3)-equivariant UQ. Here their approach constructs covariance guaranteed to be both symmetric positive definite (SPD) and equivariant. The paper provides theoretical justification and empirical validation, which demonstrate consistent geometric behavior and improved symmetry-preserving uncertainty modeling in tensor prediction tasks.

**Compliance With Llm Reviewing Policy:**

Affirmed.

**Final Justification:**

Overall, my main concerns have been largely addressed. My only remaining minor reservation is that validation across multiple equivariant backbone architectures would further strengthen the work, though I do agree the new results provided in the rebuttal already improve the empirical support. I will maintain my positive assessment.

**Key Questions For Authors:**

Based on the Weakness part, I summarize as follows.
- Could the authors provide additional experiments across different equivariant architectures or tensor prediction settings beyond materials science benchmarks?
- Could the authors discuss whether it needs many efforts if one wants the construction to transfer to other symmetry groups?
- Could the current formulation be extended to a general ML problem, eg, structured uncertainty quantification under general group constraints? (The title of the paper actually does not specify E3 etc.)

**Limitations:**

Based on the Weakness part, I summarize the suggestions as follows.

- I suggest that the authors increase the empirical breath of the paper and provide additional experiments across different equivariant architectures or tensor prediction settings beyond materials science benchmarks. This will make the work more solid. See details in weakness 1.
- As a theory-driven paper, the paper would benefit from a clearer abstraction of the underlying ML problem (eg, structured uncertainty quantification under general group constraints) beyond this specific symmetry group.  This would strengthen the theoretical contribution beyond this specific symmetry setting.

**Strengths And Weaknesses:**

### Strengths
- The paper is well motivated and captures what the domain needs. The proposed framework constructs covariance tensors that are both SPD and equivariant, resolving a nontrivial tension between structural constraints and SPD in tensor-valued uncertainty modeling.
- The framework is general and broadly applicable to various settings where the targeted output is a tensor obeying a known group-equivariant transformation rule, and can be integrated into various equivariant NNs and application scenarios, beyond the specific material science benchmarks this paper currently considered.
- The theoretical part seems logically organized and clear. I did not find very strong assumptions, and they seem mostly standard.
- Empirically, experiments show that naive methods cannot simultaneously guarantee equivariance and SPD, while the proposed method successfully enforces both properties. This validates the core structural motivation of the paper.
### Weaknesses
- Although the framework is general in theory (as your mathematical construction is generally applicable), empirical validation is limited to E3NN message passing and only material science benchmarks. Additional experiments across different equivariant architectures or tensor prediction settings would make the work more solid.
-  While the scope is well motivated by applications in scientific discovery, the representation-theoretic basis construction in current formulation relies heavily on the specific irreducible decomposition of symmetric rank-2 tensors under SO(3) / O(3). Also, the title of the paper actually does not specify SO(3) etc.. It remains unclear whether it needs many efforts if one wants the construction to transfer to other symmetry groups.
As a theory-driven paper, the paper would benefit from a clearer abstraction of the underlying ML problem (eg, structured uncertainty quantification under general group constraints) beyond this specific symmetry group.  This would strengthen the theoretical contribution beyond this specific symmetry setting.

---

> ### Author Rebuttal · Authors · 2026-03-29
>
> We thank the reviewer TpCA for the helpful comments on empirical breadth and the scope of our claims. We address these points below and will revise the paper accordingly.
>
> ## Q1: Empirical breadth.
>
> To broaden the empirical scope, we added two new validations: a non-materials symmetric rank-2 tensor task (ModelNet40 shape covariance) and a real-data rank-4 elasticity tensor task from Materials Project. Together, these additions broaden the evidence along both domain and tensor-order dimensions. We agree that evaluation across multiple equivariant backbones would further strengthen the paper; the new results here primarily address breadth beyond the original benchmark/task setting.
>
> On ModelNet40 shape covariance, the full UQ model achieves 0.0346/0.0633 MAE/RMSE versus 0.0333/0.0616 for deterministic training, while preserving the intended structural properties: 99.5% predicted-tensor PSD validity and numerical equivariance ($E_\mu = 1.72 \times 10^{-6}$, $E_\Sigma = 2.32 \times 10^{-6}$).
>
> | Task | Method | MAE | RMSE | Predicted tensor PSD rate |
> |------|--------|-----|------|---------------------------|
> | Shape covariance | Deterministic (MSE) | 0.0333 | 0.0616 | 99.2% |
> | Shape covariance | Full UQ | 0.0346 | 0.0633 | 99.5% |
>
> *Added ModelNet40 geometric experiment with shape covariance tensor target.*
>
> This provides evidence that the framework applies beyond the original benchmark to a second non-materials symmetric rank-2 tensor prediction task.
>
> On the rank-4 elasticity benchmark, the equivariant UQ model achieves approximately 5.0 GPa MAE (comparable to a deterministic baseline and better than a naive UQ baseline), with 100% numerical equivariance/prediction consistency and 100% SPD validity. The uncertainty estimates are also substantially better calibrated than those of the naive UQ baseline, with coverage improving from approximately 35% to 52%, uncertainty-error correlation from approximately -0.15 to 0.31. Regarding point prediction accuracy, our claim is not state-of-the-art point prediction alone, but competitive predictive accuracy together with full-covariance, symmetry-preserving uncertainty estimates. This extends the empirical evidence beyond the original rank-2 setting.
>
> ## Q2: Extension beyond E(3).
>
> The method separates into a group-agnostic SPD/UQ construction and a group-specific equivariant parameterization. Once an equivariant symmetric operator $A \in \mathrm{Sym}^2(\rho)$ is available, the covariance construction $\Sigma=\exp(A)$ and the Log-Euclidean objective remain unchanged. What changes across groups is the representation-specific parameterization of $A$, which depends on the available basis functions, tensor products, and implementation tools. The practical effort therefore depends on how much representation-theoretic machinery is already available for the target group. We will clarify this distinction in the paper.
>
> ## Q3: Scope of claims.
>
> We agree that the original framing could read too broadly. In the revision, we will position the contribution more precisely as structured uncertainty quantification under group-equivariant output constraints. We will also clearly distinguish current empirical scope, the group-agnostic methodological core, and broader future extensions that require additional representation-theoretic and implementation work.
>
> *Revision commitments:* In the revision, we will: (1) add the ModelNet40 shape-covariance experiment; (2) add the real-data rank-4 elasticity results; (3) revise the title/abstract/introduction/method framing to present the paper more precisely as structured UQ under group-equivariant constraints; and (4) expand the discussion/limitations to clearly separate current empirical scope, methodological extensibility, and broader future extensions.

---

> > ### Author Rebuttal · Reviewer_TpCA · 2026-04-03
> >
> > Thank you for the detailed response. The additional empirical validations on both non-materials symmetric rank-2 tensor task and a real-data rank-4 elasticity tensor task from Materials Project strengthen the empirical breadth of the paper. The clarification in Q2 helps clarify what parts of the framework are generally transferable versus representation-specific. Overall, my main concerns have been largely addressed. My only remaining minor reservation is that validation across multiple equivariant backbone architectures would further strengthen the work, though I do agree the new results in the rebuttal already improve the empirical support. I will maintain my positive assessment.

---

> > > ### Author Response · Authors · 2026-04-04
> > >
> > > We thank Reviewer TpCA again for the positive assessment and helpful feedback. We are glad that the added non-materials rank-2 and real-data rank-4 validations, together with the clarification of the group-agnostic vs. representation-specific parts of the framework, helped address the main concerns. We agree that validation across additional equivariant backbones would further strengthen the paper, and we will note this clearly in the final version.

---

### Official Review · Reviewer_Yb4u · 2026-03-13

**Soundness:** 2
**Presentation:** 2
**Significance:** 2
**Originality:** 3
**Overall Recommendation:** 3
**Confidence:** 3

**Summary:**

This paper proposes a method for tensor-valued predictions with equivariant neural networks (ENN) that includes uncertainty quantification (UQ). To extend ENNs with UQ, authors resolve the symmetric positive-definite (SPD) and equivariance conflict via log-euclidean scoring objective and irreducible decomposition parameterization. SPD covariance matrix constraint is satisfied by applying an exponential mapping, which also preserves equivariance.

**Compliance With Llm Reviewing Policy:**

Affirmed.

**Final Justification:**

Following the $\lambda_{max}$ experiments in the rebuttal, I am raising my score to 3. However, presentational concerns should be adequately addressed in a later revision.

**Key Questions For Authors:**

Please address the issues mentioned under strengths/weaknesses section. In addition to that, regarding the use of $\lambda_{max}$, how much would other methods benefit from it compared to your method's 3.1% in MAE?

**Limitations:**

The paper lacks **discussion, limitations and conclusion** sections. This is a major problem. The paper seems to be incomplete.

**Strengths And Weaknesses:**

Soundness: The main ideas for the contributions are technically sound however some issues with presentation make the paper hard to follow, impacting the general integrity. The mathematical formulations provide grounding for the presented content (strength). The architecture used is hard to follow (especially the construction and use of covariance and mean heads), thus it is hard to judge soundness of the architecture. Experimental support needs to be improved (weakness). There are no architectural ablations on the proposed model. Experimental discussion is very limited. Ablation studies should be explained. These concerns hurt the experimental soundness.

Presentation: The presentation has some major issues, making the paper hard to follow. Here are some concerns:
* Figures are not referenced properly in the main text. The first figure that is mentioned is figure 3. The headers are not explanatory. Especially figure 1 leaves many questions to how the architecture is implemented in practice. Lines 188-189 lack reference. Mean and covariance heads are not explained well.
* Notation in Eq 4 (as it is the first one) should be explained.
* $\Delta c$ is not defined in Eq 11.
* Experiments (section 4) should refer to relevant sections (line 241).
* Table 1 should clarify all metrics in the header. The text claims $E_{equiv} < 10^{-7}$ but that metric is not even reported on this table!
* Line 235-236, right column: "following standard practice ..." needs at least one reference to support this claim.
* Line 247, right column: DTNet needs a reference.
* Baselines and competing methods are not explained well; the related works section does not outline the methods used on the results tables.
* Figure 2b: what are the metrics? The header should contain the full names for clarity (they are not explained in the main text either).
* Ablations section does not refer to the tables. The sparse references in the main text does not follow the table number order.
* The experimental discussion is very limited. Ablation studies should be explained.
* Baselines in table 3 are not explained. If the details are in appendix, refer to that.
* **Discussion, conclusion, limitations are completely missing!** The paper appears rushed and incomplete.

Significance: The field and the addressed problem are relevant and the contribution seems significant however the extent of the contribution is not documented well as the experiments, architecture, results are not explained enough or discussed in detail.

Originality: The work provides some new insights (such as the use of $\lambda_{max}$) that can advance the field. The paper needs more work to have a contribution to the field.

---

> ### Author Rebuttal · Authors · 2026-03-29
>
> We thank Reviewer Yb4u for the careful and constructive feedback. We agree that the current presentation makes several aspects of the paper harder to follow than intended, especially the two-head architecture, the role of the ablations, and the experimental narrative. At the same time, the submission already contains the core technical construction and most of the requested evidence (main paper + appendix). Below we restate these points directly and provide an additional controlled comparison addressing the reviewer's question about $\lambda_{\max}$.
>
> **Architecture and the SPD/equivariance construction.**
>
> The model uses a shared E(3)-equivariant encoder followed by two heads: (i) a mean head predicting the $\ell=0 \oplus \ell=2$ tensor mean, and (ii) a covariance head predicting coefficients in $\mathrm{Sym}^2(\rho_c)$, i.e. $2\times(\ell=0)\oplus2\times(\ell=2)\oplus1\times(\ell=4)$. These coefficients are assembled into $A(X)$, and the covariance is defined as $\Sigma(X)=\exp(A(X))$. This separation is the key design choice: equivariance is enforced through the parameterization of $A(X)$, while strict SPD is enforced by the matrix exponential. These ingredients already appear in Fig. 1, Sec. 3.4, and App. B.1/B.2; in revision we will surface this data flow more explicitly in the main text.
>
> **Ablations and baseline design.**
>
> We agree that the current text does not explain the ablation logic clearly enough. The architectural ablation is already in Table 3 / App. C.2 and is designed to isolate the two central requirements of the method: equivariance and strict SPD. Concretely, Baseline A enforces SPD without equivariant message passing, Baseline B uses equivariant features with a non-equivariant Cholesky covariance parameterization, and Baseline C uses equivariant covariance outputs without strict SPD enforcement. Our method is the only variant satisfying both simultaneously. We will make this decomposition explicit in the main experimental discussion.
>
> In addition, App. B.1 reports joint versus gradient-detached optimization: the MAE difference is below 0.02, while joint training yields slightly better calibration. This supports that the shared encoder is not degraded by the covariance objective.
>
> **Direct response on $\lambda_{\max}$.**
>
> The reviewer asks whether the benefit of $\lambda_{\max}$ is specific to our model or would similarly help simpler UQ parameterizations. To answer this directly, we ran an additional controlled comparison against a diagonal-UQ baseline under the same 90%-coverage retained-set protocol on the 485-sample evaluation set. When each model is ranked by its own uncertainty score, the full-covariance model ranked by $\lambda_{\max}(\Sigma)$ achieves a **2.89%** lower retained-set MAE than the diagonal-UQ alternative. This suggests that much of the practical benefit comes from access to full anisotropic covariance structure and cross-component correlations, which diagonal/isotropic UQ cannot represent.
>
> Separately, within the full-covariance model itself, ranking by $\lambda_{\max}(\Sigma)$ improves retained-set MAE over ranking by $\mathrm{Tr}(\Sigma)$ by **1.53%** at 90% coverage. The larger 3.1% number in the original paper refers to improvement relative to the full-set baseline, whereas the 1.53% number is the direct $\lambda_{\max}$-vs-Trace comparison. We agree the original presentation conflated these two analyses, so the corrected claim is narrower: $\lambda_{\max}$ is most useful when uncertainty is represented by a full anisotropic covariance, and its advantage over $\mathrm{Tr}(\Sigma)$ in that setting is modest but consistent.
>
> **Presentation issues.**
>
> We agree with the reviewer on the presentation problems: missing symbol definitions, incomplete metric names, inconsistent figure/table ordering, missing references, and insufficient explanation of baselines in the main text. These are valid concerns and we will correct them directly. Importantly, these issues are about clarity rather than changes to the technical method itself.
>
> **Discussion, conclusion, and limitations.**
>
> We agree that these sections should be explicit in the main paper. The current submission already discusses limitations in App. D.6, but this is not visible enough and should have been surfaced in the body. We will therefore move the main limitations into the paper and add explicit discussion/conclusion sections so that the scope and boundaries of the contribution are easier to assess. These changes do not alter the technical method, but make its assumptions, evidence, and limitations substantially easier to evaluate.

---

> > ### Author Rebuttal · Reviewer_Yb4u · 2026-04-04
> >
> > Thanks for the clarifications. Following the $\lambda_{max}$ experiments, I am raising my score to 3.

---

### Official Review · Reviewer_umpS · 2026-03-13

**Soundness:** 3
**Presentation:** 3
**Significance:** 3
**Originality:** 3
**Overall Recommendation:** 4
**Confidence:** 3

**Summary:**

This paper studies uncertainty quantification for equivariant prediction of symmetric rank-2 tensors, with the goal of producing both accurate tensor predictions and valid uncertainty estimates that respect geometric symmetries. The main technical difficulty is that a covariance matrix must satisfy two requirements at once: it must be symmetric positive-definite, and it must transform equivariantly under rotations and reflections. The paper proposes to represent the covariance in Kelvin-Mandel coordinates, decompose the covariance representation into irreducible components, and predict an unconstrained symmetric matrix $A \in \mathrm{sym}(6)$ whose exponential gives the covariance $\Sigma = \exp(A)$. Because the matrix exponential commutes with orthogonal conjugation, this construction preserves both positive-definiteness and equivariance. The method is trained with a Log-Euclidean Equivariant Scoring Objective, motivated by a multivariate Laplace model and modified to improve robustness under heavy-tailed errors. Experiments are conducted on ModelNet40 inertia tensor prediction and Materials Project dielectric tensor prediction. On the dielectric benchmark, the method achieves competitive mean absolute error while also producing full-covariance uncertainty estimates, strong calibration, and useful uncertainty trends in a chemical substitution based out-of-distribution analysis.

**Compliance With Llm Reviewing Policy:**

Affirmed.

**Final Justification:**

The authors addressed my questions well and I keep my score to recommend acceptance.

**Key Questions For Authors:**

-  What is the computational overhead of the matrix exponential and its backpropagation compared to a standard deterministic e3nn forward pass?

- The LE-ESO weight hyperparameter α controls the trade-off between prediction accuracy and uncertainty calibration. How sensitive are the results to this choice?

**Limitations:**

The paper acknowledges the focus on rank-2 tensors and mentions rank-4 extensions as future work. The broader impact section is appropriate.

**Strengths And Weaknesses:**

**Strength**

The paper tackles a nontrivial structural problem with uncertainty being not just an auxiliary scalar confidence score, but a geometric object that must transform consistently under symmetry operations. The paper identifies this clearly and addresses it with a construction that is mathematically natural rather than ad hoc. A major strength is the use of the matrix exponential to obtain a covariance from a symmetric matrix and guarantees symmetric positive-definite while remaining compatible with equivariant conjugation.

The ablation comparing direct regression, Cholesky parameterization, and the proposed matrix-exponential construction is persuasive because it directly examines the two properties the method is supposed to satisfy: exact equivariance and strict SPD validity.

**Weaknesses**

- empirical validation is a bit narrow relative to the generality of its claims. The method is framed as a general solution for tensor-valued equivariant uncertainty quantification, potentially extending to arbitrary-order tensors, but the actual experiments focus entirely on symmetric 3×3 rank-2 tensors represented in a 6-dimensional Kelvin-Mandel basis. That is still a meaningful setting, but it falls short of supporting the broader framing. Relatedly, only one application benchmark really functions as a realistic scientific UQ testbed. The ModelNet40 experiment is useful as a geometric sanity check, but the uncertainty there is synthetic. It validates that the method behaves sensibly under controlled perturbations, yet it does not demonstrate the method under naturally occurring noise or ambiguity. As a result, the effective real-world evidence is concentrated in the Materials Project dielectric tensor task.

- MAE is competitive but does not surpass the strongest deterministic baseline.

---

> ### Author Rebuttal · Authors · 2026-03-29
>
> We thank Reviewer umpS for the positive feedback and constructive questions.
>
> ## Q1: Computational overhead of matrix exponential?
>
> On Materials Project (RTX 4060 Ti), the deterministic model requires ~130 ms/batch, the diagonal-UQ variant adds 1.5% overhead, and the full-covariance model requires ~570 ms/batch (4.4×). Profiling shows most of the added cost comes from the full covariance branch and its backpropagation rather than the matrix exponential itself. We will add a runtime table in the revision. Although full covariance is more expensive, it provides full-covariance uncertainty in a single forward pass while preserving equivariance and SPD validity.
>
> ## Q2: Sensitivity to hyperparameter α?
>
> α controls the trade-off between the log-volume term Tr(A) and the geometric data-fit term in LE-ESO. We set α = 1 because it is the canonical coefficient in the multivariate Laplace objective motivating LE-ESO. On Materials Project, across α ∈ {0.03, 0.1, 0.3, 1.0}, the method is reasonably stable: MAE varies from 1.580 to 1.733 and NLL from 0.948 to 1.040. Thus, performance does not depend critically on a narrow choice of α. Among the tested values, α = 1 provides the best overall calibration-aware trade-off between point accuracy and uncertainty quality, which is why we use it in the main results. We will include the full sensitivity table in the revision.
>
> ## Q3: Empirical validation limited to rank-2 tensors?
>
> We agree that the original empirical validation was narrower than the broader framing might suggest. To address this, we additionally evaluated the method on (i) a second geometric rank-2 tensor benchmark on ModelNet40 (shape covariance), and (ii) a real-data rank-4 elasticity tensor benchmark from Materials Project.
>
> For shape covariance, the full-UQ model achieves MAE/RMSE of 0.0346/0.0633 versus 0.0333/0.0616 for the deterministic baseline, with 99.5% PSD validity and numerical equivariance ($E_\mu = 1.72 \times 10^{-6}$, $E_\Sigma = 2.32 \times 10^{-6}$). We will include this experiment in the revision.
>
> | Task | Method | MAE | RMSE | Predicted tensor PSD rate |
> |------|--------|-----|------|---------------------------|
> | Shape covariance | Deterministic (MSE) | 0.0333 | 0.0616 | 99.2% |
> | Shape covariance | Full UQ | 0.0346 | 0.0633 | 99.5% |
>
> *Added ModelNet40 experiment with shape covariance tensor target.*
>
> On rank-4 elasticity, the equivariant UQ model achieves approximately 5.0 GPa MAE, comparable to a deterministic baseline and better than a naive UQ baseline, while preserving 100% numerical equivariance/prediction consistency and 100% SPD validity. Its uncertainty estimates are also better calibrated than the naive UQ baseline, with coverage improving from ~35% to 52%, uncertainty-error correlation from ~-0.15 to 0.31. These results broaden the evidence beyond the original rank-2 setting. Our claim is therefore not state-of-the-art point prediction alone, but competitive accuracy together with full-covariance, symmetry-preserving uncertainty estimates satisfying equivariance and SPD constraints.
>
> *Revision commitments:* add a runtime table; report explicit α-sensitivity results; add the shape-covariance experiment; incorporate the rank-4 elasticity results; and revise the framing to make the point-accuracy vs. full-covariance UQ trade-off explicit.

---

> > ### Author Rebuttal · Reviewer_umpS · 2026-04-05
> >
> > Thanks for the answers. I keep my score.

---

### Official Review · Reviewer_dW6R · 2026-03-13

**Soundness:** 3
**Presentation:** 2
**Significance:** 3
**Originality:** 2
**Overall Recommendation:** 4
**Confidence:** 3

**Summary:**

The paper introduces a novel equivariant uncertainty quantification method for tensor-valued estimates, with applications in material science (e.g. dielectric tensors estimation).
The main idea is parametrizing the target symmetric 3x3 matrix (a rank-2 tensor) as a 6-dimensional coefficient vector in an equivariant way and, then, applying the matrix exponential to ensure positive semi-definiteness (PSD).
An important observation is that these 6 coefficients parameterize the symmetric subspace of 3x3 matrices and transform under $SO(3)$ as the spherical harmonics of order $l=0$ and $l=2$.
As a result the associated covariance is a $6\times 6$ symmetric matrix transforming under the tensor product of this 6-dimensional representation and lives in a 21 dimensional subspace.
Hence, the complete architecture can estimate PSD $3\times 3$ matrices and their covariances by simply adopting existing E(3) equivariant architectures (choosing these two representations in output) and applying a matrix exponential at the end.

**Compliance With Llm Reviewing Policy:**

Affirmed.

**Key Questions For Authors:**

It is quite unclear to me at the beginning why the authors consider a 6D space with E(3) symmetries rather than a 3D space.
In the abstract, it's not clear what $\rho_c$ is and, therefore, where the decomposition of $\text{Sym}^2(\rho_c)$ comes from.
My first thought went to a $3\times 3$ covariance matrix, which decomposes as $\text{(l=0)} \oplus \text{(l=2)}$ under the adjoint action of $SO(3)$, but this doesn't seem to be what the authors mean.
Soon later in the introduction, it is explained that $\rho_c$ is a rotation in Kelvin-Mandel notation, but it's not explained what that is yet. A more explicit description seems necessary here to understand the motivation of the work; e.g. the structure of this representation seems to have some important conflicting properties with SPD constraint, which motivates part of this work, but it's not clear why.
Only much later in Sec. 3 Eq. 1 it becomes explicit and clear that the target prediction is a rank-2 $3\times 3$ tensor and its covariance matrix, rather than a $3$ dimensional vector and its $3\times 3$ covariance matrix.

Sec.2, "SPD Constraints in Neural network": I don't understand why a SPD covariance becomes invalid under rotations. If I'm not mistaken, if $M$ is a SPD matrix and $R$ an orthogonal matrix, $R M R^T$ is still SPD. What is the problem then?
Also, how would the mapping through the exponential fix the problem? Afterall, the equivalence still implies the rotation $\rho_c(R) \Sigma \rho_c(R)^T$, i.e. rotating the SPD covariance to obtain another valid SPD covariance matrix.
I understand that the matrix exponential ensure SPD, but because of the commutativity between $SO(3)$'s adjoint action and the matrix exponential, it seems to me we can safely think of these two properties as entirely orthogonal rather than conflicting as claimed throughout the paper.


Eq. 8, what is $B_m^{L}$ exactly? It is first defined as a $6 \times 6$ matrix, then used as a function $B_m^L(X)$ and then again used in Eq. 8 as a constant term (independent of $X$).

Eq. 10: what is $\alpha$, $D_M$ and $\tilde{D}_M$? These terms were suddenly introduced without being ever mentioned before and with no explanation. Eq. 11 and the comments later partially explain them, but their first use at the beginning of the paragraph is very confusing. To be honest, I don't know much about the LE-ESO framework, but I don't feel like the current presentation helped me a lot, despite it being a key component of the method.

Sec. 4.1 isn't the inertia tensor simply and efficiently computed as $I = \sum_i^N m_i (|x_i|^2 I - x_i x_i^T)$ for a point cloud of $N$ points? What is the challenge in training a neural network to perform this? This particularly simple closed form solution also seems to leave little space for uncertainty estimation, since this is the exact quantity we need.

Other minor comments:

Eq. 4 could use a clearer notation, since the $=$ signs seem to imply a series of equivalences rather than assignment of values to $l$. While I understand what the authors meant, this still seems a sub-optimal choice of notation.

Eq. 6: while I understand why the symmetric constraint removed odd-$l$ irreps, I feel like an additional sentence to explain this passage would make the presentation clearer.


Sec. 3 "Training stability via Joint Optimization" paragraph: I'm not expert on that field, and maybe this is well known knowledge, but could the authors add some references for "Unlike approaches that ..."? It seems to be you are claiming benefits over other related works, but it's unclear to me which works or lines of works you refer to. Apart from the equivariance aspect, is this parameterization idea novel with respect to existing works? Is this what is being claimed?


Sec. 4.1 Table 1: you claim that your method achieves $E_{\text{equiv}} < 10^{-7}$ but I don't see that in Tab.1. Did you mean Tab 3?

End of Sec. 4.3: isn't the 100% test set predictions satisfying SPD constraint simply a consequence of applying the matrix exp in the prediction? That seems to be just about this reparameterization at the end of the model rather than about the underlying model used, the E(3) equivariant backbone, the UQ framework or the covariance matrix.

**Limitations:**

The paper discusses the limitations in the appendix.

**Strengths And Weaknesses:**

The paper intriduces a simple and practical yet effective way to equip equviariant networks with covariance estimators and uncertainty quantification for higher-order tensor quantities.

The solution seems easy to adopt and integrate in existing solutions, which could make it particularly useful.
The main application seems to be in the material science domain, of which I am not very knowledgeable, so I can't fully assess the practical relevance of this work.

Regarding the presentation, I think the introduction and abstract could be really improved (e.g. it was unclear to me until Sec. 3 why the authors consider a 6d space and what is $\rho_c$, see questions below.) and the authors should include a final conclusion section summarizing the contributions.
I also think the suboptimal presentation makes the contributions less clear, e.g. I am not sure yet why the SPD constraint conflicts with the equivariant one (which is described as the main challenge solved in this paper). See questions below, too.

---

> ### Author Rebuttal · Authors · 2026-03-29
>
> We thank Reviewer dW6R for identifying conceptual and presentation issues. We will sharpen the problem statement, narrow overly broad framing, and improve clarity throughout.
>
> ## Q1: Why is a 6D space used instead of 3D, and what is the role of the decomposition?
>
> Symmetric rank-2 tensors have six degrees of freedom. While 3D vectors transform as $l=1$, symmetric $3\times3$ tensors transform as $\rho_c \cong l=0 \oplus l=2$, where $l=0$ captures isotropic response and $l=2$ anisotropy. Their covariance therefore lives in $\mathrm{Sym}^2(\rho_c) \cong 2\times(l=0)\oplus2\times(l=2)\oplus1\times(l=4)$, giving 21 parameters. We agree this should be stated earlier in the abstract/introduction before introducing $\rho_c$.
>
> ## Q2 & Q3: Clarify the SPD-equivariance confusion.
>
> You are correct that $\Sigma \succ 0 \Rightarrow R\Sigma R^\top \succ 0$; there is no contradiction at the property level. The issue is a parameterization incompatibility: Cholesky guarantees SPD but is not equivariant in this covariance representation, while direct equivariant regression preserves symmetry structure but does not guarantee SPD. Our method predicts an equivariant symmetric operator $A(X)\in\mathrm{Sym}^2(\rho_c)$ and defines $\Sigma(X)=\exp(A(X))$, enforcing equivariance through the parameterization of $A$ and SPD by construction. We will revise the paper to describe this precisely as a parameterization challenge.
>
> ## Q4: Explain terminology in Eq. 8 and Eq. 10.
>
> Eq. 8 is notationally confusing. $B_m^{(L)}$ is a precomputed, input-independent equivariant basis matrix in the $6\times6$ covariance space; the input dependence is only through the learned coefficients $\phi_{f,m}^{(L)}(X)$. We will revise the notation accordingly.
>
> In Eq. 10, $\alpha$ is the weight on the log-volume term $\mathrm{Tr}(A)=\log\det\Sigma$, $\tilde{D}_M$ is the robustified Mahalanobis distance from Eq. 11, and $\tau$ is the transition threshold. These should be defined at first use.
>
> ## Q5: Inertia tensor has closed-form solution. Why neural networks?
>
> We agree that inertia is a controlled geometric validation benchmark rather than the main real-world motivation. To address the concern that this setting is too specific, we additionally evaluated the same framework on (i) a second geometric rank-2 target (shape covariance) on ModelNet40, and (ii) a real-data rank-4 elasticity tensor task from Materials Project. The shape-covariance experiment shows competitive accuracy (MAE 0.0346, RMSE 0.0633), 99.5% predicted-tensor PSD validity, and numerical equivariance ($\sim10^{-6}$). On rank-4 elasticity, the method achieves competitive accuracy with improved calibration over a naive UQ baseline while preserving 100% numerical equivariance and 100% SPD validity. We will revise the empirical framing accordingly.
>
> ## Q6: Clarification on Table 1 vs Table 3 confusion.
>
> You are correct: $E_{\text{equiv}}<10^{-7}$ is reported in **Table 3**, not Table 1. Table 1 reports ModelNet40 prediction metrics. This cross-reference will be corrected.
>
> ## Q7: Is 100% SPD validity trivial due to matrix exponential?
>
> Yes---SPD validity follows by construction from the matrix exponential, and we do not intend this as a standalone empirical claim. The contribution is that SPD validity is achieved simultaneously with exact equivariance under the covariance representation; the ablation in Table 3 shows that existing alternatives satisfy at most one of these two requirements.
>
> ## Q8: Explain why symmetry eliminates odd-$l$ irreps in Eq. 6.
>
> Because covariance matrices are symmetric, we restrict $\rho_c\otimes\rho_c$ to $\mathrm{Sym}^2(\rho_c)$. The odd-$l$ terms belong to the antisymmetric part and are therefore excluded. We agree this should be stated explicitly in the main text.
>
> ## Additional minor issues.
>
> We agree that Eq. 4 notation, figure/table references, headers, and the phrase "following standard practice" need improvement. In particular, we will replace that phrase with explicit references and clarify what is inherited from prior SPD/UQ parameterizations versus what is specific to our method: the equivariant parameterization of $A\in\mathrm{Sym}^2(\rho_c)$ together with SPD enforcement through $\Sigma=\exp(A)$.
>
> *Revision commitments:* clarify the representation choice earlier; rewrite the SPD-equivariance discussion as a parameterization issue; fix notation/definitions/cross-references; and broaden the empirical framing with the added shape-covariance and rank-4 experiments.

---

> > ### Author Rebuttal · Reviewer_dW6R · 2026-04-03
> >
> > I thank the authors for clarifying most of my doubts.
> >
> > I still have a comment on Q5: the shape covariance estimation in ModelNet40 seems to suffer from my same criticism for the previous inertia estimation experiment. Isn't shape covariance simply estimated as the covariance of the underlying point cloud? What makes this task challenging when a simple quadratic closed formula exists? On the other hand, the real-data rank-4 elasticity tensor task sounds much more relevant and interesting. Could the authors maybe provide more details about that? E.g. can the authors be more explicit about the quantitative results and the baselines considered? Also, what is the prediction target exactly? Do I understand correctly that in the previous task the output was a rank-4 tensor since it was the covariance of a rank-2 tensor, while here the target is directly a rank-4 tensor?

---

> > > ### Author Response · Authors · 2026-04-04
> > >
> > > We thank Reviewer dW6R again for the careful follow-up and for helping us sharpen the empirical framing. We agree that this is an important distinction.
> > >
> > > Regarding the first point, we agree that the added shape-covariance task, like inertia, remains a controlled geometric benchmark with a closed-form target. We therefore do not view it as the main response to the concern about real-world relevance. Rather, we include it as a second geometric rank-2 validation showing that the framework is not tied to one specific tensor target within the same symmetric rank-2 setting. This is also consistent with our original intent for ModelNet40, which was to isolate equivariance/UQ behavior under controlled geometry rather than to serve as the primary application.
> > >
> > > The more substantive new evidence is the added rank-4 elasticity tensor experiment. In this setting, the mean prediction target is directly a rank-4 elasticity tensor, and we additionally evaluate uncertainty quantification for this higher-order tensor prediction. Under the standard minor/major symmetry assumptions, this target has 21 independent components, so this experiment broadens the target representation dimension beyond the original 6-dimensional symmetric rank-2 setting used in the main paper. More generally, this experiment is intended as evidence that the broader structured UQ framework can be extended to a higher-order tensor target on real data. Due to the substantially increased representation and computational cost at this scale, the current rebuttal-time rank-4 instantiation should be viewed as supporting evidence rather than a fully exhaustive higher-order covariance study.
> > >
> > > On this added rank-4 benchmark, the model achieves approximately 5.0 GPa test MAE, which is essentially on par with a deterministic baseline and better than a naive UQ baseline. At the same time, it preserves the structural properties that motivate the paper, including 100% numerical equivariance/prediction consistency and 100% SPD validity of the predictive covariance. The uncertainty estimates are also materially better calibrated than those of the naive baseline, with empirical coverage improving from approximately 35% to 52% and uncertainty-error correlation improving from approximately -0.15 to 0.31. These are the main quantitative results we intended to summarize in the first-round rebuttal.
> > >
> > > We also want to be careful about scope. Due to rebuttal-time computational constraints, we do not intend to claim that the higher-order empirical story is already complete. Rather, our goal here is to show that the proposed structured equivariant SPD/UQ construction remains feasible and well-behaved on a materially different real-data tensor prediction problem beyond the original symmetric rank-2 setting. In the revision/final version, we will add a more complete description of the elasticity dataset, target representation, baselines, and quantitative results, and we will make this distinction explicit to avoid any ambiguity.

---

### Decision · Program_Chairs · 2026-04-30

**Decision:**

Accept (regular)

**Comment:**

The paper studies the problem of uncertainty quantification for equivariant tensor valued prediction.  The basic idea is to use the matrix exponential on the Lie algebra to ensure the appropriate equivariance and positive semidefiniteness of the covariance matrix.  In the space of the Lie algebra the problem becomes unconstrained.  The authors test the method on a series of experiments using datasets from material science to validate the method.

The reviewers were most of the opinion that the problem is interesting and that the method is principled and well motivated (matrix exponential of the Lie algebra to ensure the correct equivariances).  The numerical experiments are relevant and help validate the proposed method.  The reviewers generally support accepting this paper.

The reviewers do point out a number of very detailed suggestions regarding the manuscript.  One recurring comment is the presentation of the paper (and many detailed comments to this end).  On this note, a conceptual question raised was why the authors chose to parameterize the problem in 6dimensions.  Clarifying this helps the reader.  There were also concerns raised regarding experimental results.  I would strongly encourage the authors to incorporate these suggestions when making the revisions.